# SwingArena: Adversarial Programming Arena for Long-context GitHub Issue Solving

**Wendong Xu**[1*], **Jing Xiong**[1*], **Chenyang Zhao**[2,10*], **Qiujiang Chen**[10], **Haoran Wang**[3],
**Hui Shen**[4], **Zhongwei Wan**[5], **Jianbo Dai**[6], **Taiqiang Wu**[1], **He Xiao**[1], **Chaofan Tao**[1],
**Z. Morley Mao**[4], **Ying Sheng**[10], **Zhijiang Guo**[9], **Hongxia Yang**[8], **Bei Yu**[7],
**Lingpeng Kong**[1], **Quanquan Gu**[2], **Ngai Wong**[1]

[1]The University of Hong Kong    [2]University of California, Los Angeles    [3]Tsinghua University
[4]University of Michigan    [5]The Ohio State University    [6]University of Edinburgh
[7]The Chinese University of Hong Kong    [8]The Hong Kong Polytechnic University
[9]Hong Kong University of Science and Technology (Guangzhou)    [10] LMSYS Org

🌐 **Website**    ⭘ **GitHub**    🤗 **Huggingface**

## ABSTRACT

We present SWINGARENA, an adversarial evaluation framework for Large Language Models (LLMs) that approximates real-world software development workflows. Unlike traditional static benchmarks, SWINGARENA models the collaborative process of software iteration by pairing LLMs as *submitters*, who generate patches, and *reviewers*, who create test cases and verify the patches through continuous integration (CI) pipelines. To support these interactive evaluations, we introduce a retrieval-augmented code generation (RACG) module that handles long-context challenges by providing relevant code snippets from large codebases across multiple programming languages (C++, Python, Rust, and Go). Our adversarial evaluation can surface limitations that are often overlooked by traditional evaluation settings. Our experiments, using over 400 high-quality real-world GitHub issues selected from a pool of 2,300 issues, indicate differing behavioral tendencies across models in patch generation versus validation. SWINGARENA offers a scalable and extensible approach to evaluating LLMs in CI-driven software development settings.

## 1 INTRODUCTION

Large Language Models (LLMs) have become potent accelerators of software development (Chen et al., 2021; Li et al., 2023; Lozhkov et al., 2024), capable of synthesizing high-quality code snippets, automatically detecting and repairing defects, and providing interactive guidance throughout the development cycle. To assess these three capabilities—code-generation fidelity, automated debugging, and conversational assistance—a benchmark must evaluate each dimension individually and in concert. Existing suites including HumanEval (Chen et al., 2021) and MBPP (Austin et al., 2021) focus on the functional correctness of concise, self-contained snippets, offering a valuable first glance at model proficiency. Nevertheless, their narrow scope fails to capture the richer, iterative workflows that typify modern software engineering. Recent efforts including SWE-Bench (Jimenez et al., 2023) move toward greater realism by grounding tasks in genuine GitHub issues and repositories. Nevertheless, they typically rely on static or partially simulated contexts like single unit tests, omitting the critical role of the full Continuous Integration (CI) pipeline and its automated safeguards that define professional development. Moreover, these benchmarks tend to assume a one-shot coding paradigm, whereas industrial software engineering is inherently iterative: debugging, testing, and incremental refinement are the norm, not the exception.

---

    * These authors contributed equally.

In real-world software development, coding tasks involve collaborative, iterative workflows with complex project requirements and automated systems. For example, a common scenario involves a contributor submitting a pull request (PR) to a large open-source project, where a CI pipeline—often implemented with GitHub Actions—automatically builds up validation environments, runs unit tests, enforces style guides, executes linters[1], and validates compatibility with the existing codebase. Any failure triggers an iterative dialogue between contributor and reviewers, where comments are exchanged and resolved, patches are applied, and checks are rerun until all pass, allowing the PR to be approved and merged into the upstream codebase. This submitter–reviewer loop epitomizes modern collaborative workflows, yet a robust evaluation framework for such interactions in LLMs is still lacking, leaving a critical aspect of real-world practice largely unaddressed by current benchmarks.

To meet the demands of this real-world process, conventional benchmarks (Zan et al., 2024; 2025; Rashid et al., 2025; Aleithan et al., 2024) often fall short by focusing only on basic questions like "Does the code pass a unit test?"—a threshold that is far beneath the expectations of professional software development. A meaningful evaluation must instead ask, "Can the model submit code that is valid, compliant, and able to pass a full CI pipeline and peer review?"

Current evaluation practices suffer from three critical blind spots. First, static benchmarks use fixed, predictable challenges that fail to capture the dynamic, adversarial nature of real software development where patches face adaptive scrutiny. Second, they evaluate single-agent performance in isolation, missing collaborative interactions and role-switching dynamics essential to modern software engineering workflows. Third, they focus narrowly on functional correctness while ignoring comprehensive quality gates that determine real-world success.

LLMs must overcome one of the core challenges in software engineering: effectively analyzing and interpreting over long contexts in code repositories. Realistic CI tasks involve sprawling code-bases where essential information is scattered across thousands of lines and multiple files. To enable fair evaluation across diverse model architectures and context window sizes, we implement a Retrieval-Augmented Code Generation (RACG) system that provides standardized context access using established retrieval components. This helps ensure models receive comparable context—useful for isolating the effects of our adversarial evaluation protocol from retrieval artifacts.

Thus, we introduce SWINGARENA, an adversarial evaluation framework that operationalizes full CI workflows as dynamic testing arenas for LLMs. Our contributions are:

- A standardized adversarial CI evaluation protocol that pairs a submitter and a reviewer, executed on repository-native workflows (PK-style dual-role evaluation with role switching and clear scoring).
- A multi-language long-context retrieval pipeline (RACG) combining syntax-aware chunking, dense reranking, and token-budget–aware packing across C++, Python, Rust, and Go, positioned as a strong baseline to support SwingArena rather than a standalone algorithmic contribution.
- A curated, CI-grounded dataset of 2,300 real GitHub issues with solutions (4 languages), including 400 evaluation instances (100 per language) and a 100-sample ablation split, with scripts to reproduce retrieval and CI execution.

We evaluate SWINGARENA across multiple state-of-the-art proprietary and open-source models, including GPT-4o, Claude-3.5, Gemini-2.0, DeepSeek-V3, and several open-source alternatives, demonstrating the framework's broad applicability and revealing distinct behavioral patterns across different model architectures. Our experiments reveal that while some models excel at aggressive patch generation, others prioritize correctness and CI stability, highlighting the nuanced trade-offs that emerge in realistic software engineering scenarios.

We adopt an adversarial CI protocol rather than static tests to cover repository-native gates (style, security, coverage) and the interaction between patching and testing that fixed suites under-report. This design enables us to measure end-to-end success on real pipelines and to quantify stability, reviewer effects, and RACG's contribution under context limits across models and languages.

---

[1] A linter is a tool that automatically analyzes code for grammar errors, style issues, and potential bugs to ensure consistency and quality.

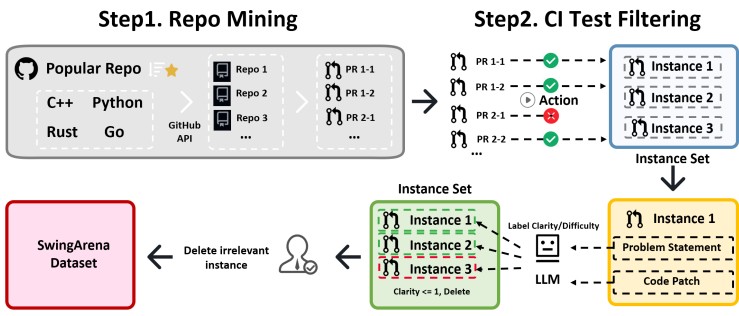

Figure 1: Overview of SWINGARENA data construction pipeline, including repository collection, pull request extraction, task instance creation, quality filtering, and multiple CI-based validation.

## 2 RELATED WORK

### 2.1 BENCHMARKS FOR EVALUATING REAL-WORLD SOFTWARE ENGINEERING

Real-world software engineering is complex, requiring nuanced reasoning over large, evolving codebases. While LLM-based agents have made significant progress (Yang et al., 2024; Zhang et al., 2024; Xia et al., 2024), evaluating the quality of generated code remains challenging (Wang et al., 2025a). SWE-Bench (Jimenez et al., 2023) takes an important step toward realism by using GitHub issues, pull requests, and unit tests grounded in real repositories. However, it is limited to Python, focuses only on unit test success, and includes noisy or weakly aligned test cases. Recent multi-language extensions (Zan et al., 2024; 2025; Rashid et al., 2025; Aleithan et al., 2024) often require manual Docker setup, hindering automation and preventing the use of stricter CI-based arenas to evaluate LLMs under realistic development workflows. How to integrate real-world workflows of multi-language code development, submission, and review into evaluation remains an open problem.

### 2.2 CODE EVALUATION

Evaluating the effectiveness of code-generation agents remains a challenge (Wang et al., 2025c). Prior work explores diverse assessment strategies (Evtikhiev et al., 2023; Yetistiren et al., 2022; Siddiq et al., 2024), but most focus on function-level correctness (Mündler et al., 2024; Zhuge et al., 2024; Liu et al., 2024b; Dunsin, 2025) and overlook repo-level effects. We push evaluation to the software-wide scale by embedding generated patches into a CI pipeline, allowing us to observe their impact on build stability, regression risk, and interactions across the codebase. Current benchmarks (Jain et al., 2024; Wang et al., 2025b) still lack the ability to stress-test models in this way, as they do not incorporate an adversarial agent that detects corner cases and synthesizes new unit tests.

### 2.3 RETRIEVAL-AUGMENTED GENERATION

The context window limitations of LLMs present a major bottleneck for handling large-scale codebases. Retrieval-augmented generation (RAG) techniques (Jimenez et al., 2023; Xie et al., 2025; Yang et al., 2024) have emerged as a practical solution. While advanced code retrieval methods explore structured representations like code graphs or abstract syntax trees for more semantic understanding, many current approaches still rely on lexical methods like BM25 (Robertson et al., 2009) for document and function name retrieval, without incorporating static code analysis or fine-grained code structure understanding. This often hampers performance on tasks that require identifying relevant functional code blocks from complex repositories (Feng et al., 2020; Parvez et al., 2021; Zhang et al., 2023; Xia et al., 2024). For SWINGARENA, we opted for a robust and language-agnostic RAG pipeline to serve as a strong, reproducible baseline, acknowledging that more sophisticated, language-specific retrieval strategies represent a promising direction for future work.

## 3 SWINGARENA

## 3.1 DATA CONSTRUCTION

This section details the data construction pipeline for SWINGARENA. The pipeline comprises several stages: Repository Mining, CI Test Filtering, LLM Filtering, and Expert Filtering. An overview of this process is presented in Figure 1.

**Repository Mining** We identify high-quality repositories via the GitHub API, prioritizing those with high popularity including star count as a proxy for code quality and community validation. We assume that patches and unit tests merged into such repositories have typically undergone extensive expert review, increasing the likelihood of correctness. For each selected repository, we collect metadata including license, forks, and activity, and clone those meeting our predefined criteria. From these, we extract real-world PRs linked to their corresponding issues, along with code diffs, patch content, and metadata. We then convert raw PRs data into structured benchmark instances by extracting problem statements from PR descriptions and issues, gathering associated patches and available test cases. Each task is enriched with contextual metadata including repo ID, base commit SHA, patch, timestamp, and CI lists, enabling realistic and fine-grained evaluation of LLM performance. We release a curated, CI-grounded dataset of 2,300 (issue, PR) pairs and provide 400 evaluation instances (100 per language) plus a 100-sample ablation split. We include license-aware distribution and scripts for reproducible retrieval and CI execution. We focus on four programming languages—Rust, Go, C++, and Python—based on their prevalence in open-source repositories and the maturity of their associated CI ecosystems. These languages dominate most large-scale codebases and account for a substantial portion of real-world software development activity.

**CI Test Filtering** After mining repositories, we integrate corresponding CI configurations including GitHub Actions and Travis CI[2] for each PR to fully replicate the real-world end-to-end software development process, including each repository's testing and build requirements. We retain only instances that pass all CI checks, ensuring they meet project-specific quality standards. Instances with verified test coverage are prioritized. This step ensures that the benchmark consists of code changes that correspond with real-world, automated validation pipelines. By integrating CI, our pipeline enforces strict validation, creating our original instances pool that mirrors the constraints and practices of professional software development.

**LLM Filtering** To improve dataset quality and balance instances' difficulty, we employ LLM-as-a-Judge (Zheng et al., 2023) to assess the clarity of each pull request's problem statement and estimate its overall difficulty. The LLM (Grok-3-beta (xAI, 2024)) is required to provide reasons for its clarity and difficulty assessments, ensuring each evaluation is supported by a rationale for the sake of final expert verification. This process enables us to categorize tasks by complexity, creating a structured and balanced benchmark.

**Expert Filtering** Following LLM-as-a-Judge, human experts finally reviewed and calibrated LLM-generated assessments. Annotators examine the clarity and difficulty scores, along with the model's rationales, and either confirm or correct them. If the model's justification is unclear or misaligned with the problem, human experts intervene to ensure consistent and accurate labeling. This step mitigates LLM hallucinations and filters out low-quality problem statements, incoherent instances, or those with unreachable expired multi-modal attachment. After all the data construction process, the final data statistics of SWINGARENA can be found in Appendix B.

## 3.2 ARENA

In this section, we introduce SWINGARENA, an adversarial evaluation framework that facilitates direct competition against models on a set of programming tasks. Unlike traditional benchmarks focused on static code snippet completion or single unit tests, SWINGARENA creates a dynamic, interactive environment that simulates real-world software development workflows. Specifically, the framework emulates the process of submitting and reviewing code changes: one model assumes the role of a *submitter*, proposing a candidate solution, while another acts as a *reviewer*, critiquing the submission and—crucially—generating additional test cases to expose potential flaws or edge cases. This interactive setup accurately mirrors human software collaboration and enables richer evaluation along multiple dimensions, including reasoning depth, code robustness, and collaborative capability.

---

[2]Travis CI is a continuous integration service that automates testing and deployment of code changes in software projects.

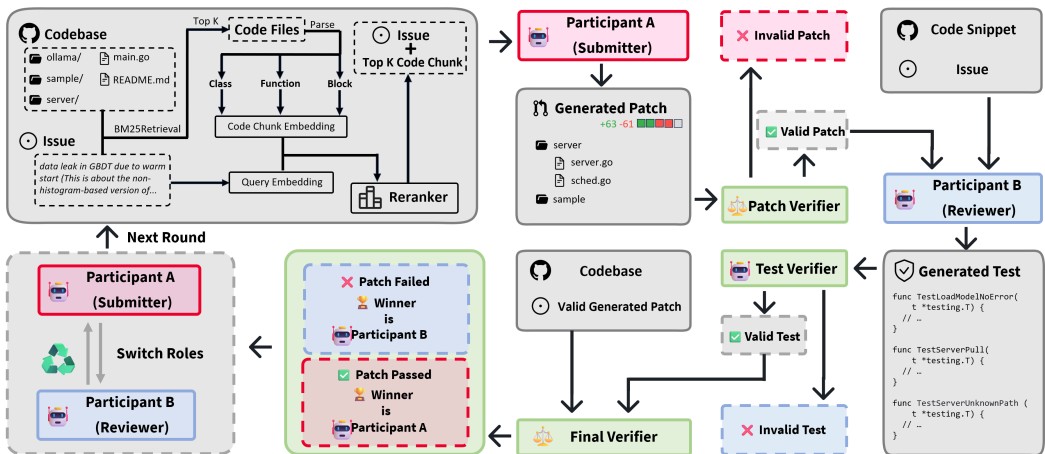

Figure 2: Illustration of the SWINGARENA adversarial evaluation framework. The framework simulates an adversarial software engineering workflow between two agents—a *submitter* and a *reviewer*—who alternate roles and iteratively refine their solutions based on CI feedback.

**Interactive Environment** SWINGARENA employs a modular, multi-agent interactive environment in which language models act as software engineers. Each model receives the same task description and access to a shared codebase but may take different approaches to generate patches and unit test cases. Our end-to-end environment includes components for retrieving relevant code, extracting and ranking code chunks, and generating code patches. Code modifications are validated using real-world CI pipelines, ensuring that models produce solutions that compile, run, and pass automated tests. This design takes a significant step toward aligning model evaluation with real-world software engineering scenarios.

**Battle Protocol** We denote a single round of adversarial patch and test case generation, evaluation, and scoring between the submitter and reviewer as a battle. Given a problem statement with a buggy program and its description, the submitter generates a patch to fix the issue, while the reviewer creates a test case to challenge the patch's correctness. A CI pipeline evaluates both: the patch is checked for compilation and correctness, and the test case is assessed for its ability to expose flaws. The submitter receives a score of +1 for patches passing all tests (including the reviewer's), or -1 for any failure. The reviewer receives a score of +1 if their test fails the submitter's patch, revealing a fault, and -1 if it fails the golden human patch. Models alternate roles across multiple rounds with CI feedback for iterative refinement, simulating dynamic software development.

**Verification** SWINGARENA integrates real-world CI workflows into the evaluation process as a verification mechanism. For each repository, existing CI pipelines—including GitHub Actions—are executed locally within container environments, preserving the exact logic, dependencies, and toolchains used by human developers. Language-specific execution logic is supported when needed including Rust's `cargo` system[3], ensuring accurate and faithful testing. All evaluations run inside isolated Docker containers to guarantee reproducibility and avoid cross-task contamination. Given that different repositories require different runtime environments, developers can mitigate environment-related inconsistencies by supplying configuration files including `Dockerfile`, `.yaml`, or `environment.yml` to automatically construct test environments in containers for consistent and automated verification.

**Evaluation** SWINGARENA evaluates models through adversarial interactions between two agents—the patch generator (the *submitter*) and the test case generator (the *reviewer*)—executed within real-world CI workflows. Each task begins by validating the baseline and golden human patch via CI to establish correctness references. In each round, the *submitter* proposes a patch, compared against the golden human fix; incorrect patches incur penalties. The *reviewer* then generates a unit test intended to expose faults. The patch and test are jointly applied and verified through CI: passing tests reward the *submitter*, and failing ones reward the *reviewer*. Both models take turns acting as the submitter and reviewer across rounds, ensuring a balanced assessment of their patch and test generation capabilities. Evaluation metrics include binary task success, win rates, and Best@k, with

---

[3]`cargo` is Rust's build tool and package manager, used to compile code, manage dependencies, and run tests.

CI logs and model outputs aggregated into structured reports.

**Reviewer Test Quality Gates** To control evaluation variance and prevent exploitative behavior, reviewer-generated tests must: compile and pass when applied to the golden patch; refrain from modifying production code or existing tests; limit edits in any new test file to a bounded number of lines; avoid sources of nondeterminism; and conform to repository linting and style guidelines. Any violation results in automatic test rejection and forfeiture of the reviewer's reward. See Appendix D for more details.

### 3.3 RETRIEVAL-AUGMENTED CODE GENERATION

Real-world programming tasks often involve reasoning over large, multi-file codebases where relevant context is scattered across thousands of lines. While prior work (Xia et al., 2024; Xie et al., 2025) has tried to address long-context challenges (mainly in Python) via AST-based parsing (Foundation, 2023), these approaches typically (i) emphasize a single language, (ii) lack a *context packing policy* under strict token budgets, and (iii) do not integrate with adversarial CI evaluation. To address these limitations, SWINGARENA introduces a multi-language Retrieval-Augmented Code Generation (**RACG**) framework that combines static code analysis and dense retrieval to efficiently extract, rank, and *pack* relevant code snippets to the model.

**File Retriever** During the file retrieval stage, SWINGARENA employs a coarse-to-fine-grained filtering step using a lightweight `FileRetriever`. This component utilizes a classical sparse retrieval method, BM25, to rank source files based on lexical similarity to the problem description, treating the problem as a query and each file as a document. This step prunes irrelevant files early, narrowing the search space and boosting efficiency for subsequent dense retrieval. Only the top-$k$ most relevant files, as ranked by BM25, are forwarded to the `CodeChunker` for decomposition into semantically meaningful code chunks. This hierarchical retrieval pipeline—from file-level sparse matching to chunk-level dense reranking—enables SWINGARENA to scale effectively to large codebases while preserving high precision in final context selection.

**CodeChunker** SWINGARENA uses a hierarchical, syntax-aware chunking strategy via `CodeChunker` to decompose codebases into semantically meaningful units including functions, classes, and blocks, preserving structural integrity and improving retrieval precision. It supports multiple programming languages—including Rust, Python, C++ and Go—through language-specific parsing rules. When parsing is unavailable or impractical, the system falls back to regex-based heuristics tailored to each language, ensuring robustness across diverse codebases.

**CodeReranker** Given the large number of candidate chunks in real-world codebases, SWINGARENA employs `CodeReranker` to prioritize content effectively within the model's context window. It uses CodeBERT (Feng et al., 2020) to encode both the problem statement and code chunks into dense vectors, ranking them by cosine similarity to identify the most relevant segments. Beyond naive top-k selection, we incorporate (i) language-aware tie-breaking (favoring definitions over usages), (ii) proximity bias (neighboring chunks near an already selected target receive a small boost), and (iii) de-duplication across files to avoid redundant inclusions.

**Token Budget-Aware Context Management** To optimize token usage under strict token limits, SWINGARENA employs a dynamic token budgeting mechanism that incrementally selects and packs code chunks until the token threshold is reached. Crucially, it adapts chunk granularity based on available context window size—favoring coarser chunks when space permits and switching to finer-grained chunks as the budget tightens. Each included chunk is enriched with metadata (file path, symbol kind, function/class name, line range). This policy is *deterministic* given the retrieval scores and budget, improving reproducibility across runs.

**Variance Control in the Adversarial Arena** The adversarial setting introduces interaction-induced variance. We bound variance via: (i) fixed system and user prompts; (ii) a capped number of rounds and retries; (iii) temperature=0 decoding in all primary evaluations, using controlled higher-temperature sampling only in the scaling-law study; (iv) unified CI recipes executed via `act` with pinned images; and (v) fixed random seeds for retrieval and any sampling components. These choices make our outcomes reproducible while retaining the benefits of interactive stress.

**Battle Protocol** We denote a single round of adversarial patch and test case generation, evaluation, and scoring between the submitter and reviewer as a battle. Given a problem statement with a buggy program and its description, the submitter generates a patch to fix the issue. Concurrently, the reviewer creates a test case designed not merely to validate, but to strategically challenge the patch's correctness by probing for edge cases and potential weaknesses. To foster this adversarial

nature, the reviewer is provided with contextual hints including which parts of the code were most changed by the patch, and is prompted to design tests that specifically target the logic of the fix. A CI pipeline evaluates both: the patch is checked for compilation and correctness, and the test case is assessed for its ability to expose flaws. The submitter receives a score of +1 for patches passing all tests (including the reviewer's), or -1 for any failure. The reviewer receives a score of +1 if their test fails the submitter's patch, revealing a fault, and -1 if it fails the golden human patch. Models alternate roles across multiple rounds with CI feedback for iterative refinement, simulating dynamic software development.

# 4 EXPERIMENTS

## 4.1 EXPERIMENTAL SETUP

**Baselines** We evaluate our method against several strong models: GPT-4o (Achiam et al., 2023), a general-purpose LLM optimized for multimodal reasoning; Claude-3.5 (Anthropic, 2024), an instruction-tuned model for code reasoning; Gemini-2.0 (Google, 2023), a multimodal model with robust programming capabilities; and DeepSeek-V3 (Liu et al., 2024a), a code-augmented model trained on large-scale software repositories. Besides, we use Qwen2.5-Coder-7B-Instruct (Hui et al., 2024) to do ablation studies.

**Data Division** We collect over 2,300 issues with corresponding solutions from GitHub, covering Rust, Python, Go, and C++. This dataset is divided into two parts: 400 high-quality samples filtered from the full set (100 per language) are used for evaluation, while the remaining data are reserved for future community-driven training. To comprehensively evaluate both the *submitter* and the *reviewer* across programming languages while ensuring experimental efficiency, we design two settings: (1) a battle scenario using the 400 selected samples, and (2) an ablation experiment using 100 samples (25 high-quality random samples from each language).

**Metrics** Let $\mathcal{T}$ denote the set of tasks and $k$ the number of independent attempts per task (when applicable). We formalize the following metrics.
**Best@$k$**: a task is counted as solved if at least one of $k$ independent generations succeeds. Formally, $\text{Best@}k = \frac{1}{|\mathcal{T}|} \sum_{t \in \mathcal{T}} \mathbf{1}\{\exists i \leq k : \text{success}(t, i)\}$.
**CI pass rate**: we report submitter-side and reviewer-side CI check pass rates averaged over tasks. Submitter CI Pass Rate (SPR) averages, for each task, the fraction of submitter-side checks passed by the generated patch (excluding reviewer tests), then averages across tasks. Reviewer CI Pass Rate (RPR) analogously averages the fraction of reviewer-generated tests that pass against the golden patch.
Formally, let $\mathcal{C}_{\text{sub}}(t)$ denote submitter-side checks for task $t$ and $\mathcal{C}_{\text{rev}}(t)$ reviewer-side checks. Then

$$\text{SPR} = \frac{1}{|\mathcal{T}|} \sum_{t \in \mathcal{T}} \frac{1}{|\mathcal{C}_{\text{sub}}(t)|} \sum_{c \in \mathcal{C}_{\text{sub}}(t)} \mathbf{1}\{\text{pass}(t, c)\}, \quad \text{RPR} = \frac{1}{|\mathcal{T}|} \sum_{t \in \mathcal{T}} \frac{1}{|\mathcal{C}_{\text{rev}}(t)|} \sum_{c \in \mathcal{C}_{\text{rev}}(t)} \mathbf{1}\{\text{pass}(t, c)\}.$$

**Win Rate**: the fraction of battles whose final outcome is that the submitter's patch passes all CI checks (including reviewer tests) and agrees with the golden fix. Note that Win Rate is *adversarial*: higher values may also indicate weaker reviewer tests, so it should be interpreted together with SPR/RPR.

**Implementation Details** For both patch and test case generation in the arena evaluation, we employ carefully crafted system and user prompts. The system prompt instructs the model to behave as a senior software engineer or test automation expert, while the user prompt provides the issue description, relevant code snippets, and contextual metadata. All model outputs are required to follow a structured JSON schema to enable automated downstream evaluation.

We set the generation temperature to 0 to ensure deterministic outputs. The maximum number of generated tokens is configured based on the model's context window and the specific requirements of each task. For RACG, we limit the number of retrieved files to 5 and allow a maximum of 16 code chunks per query, where each chunk corresponds to a syntactic code chunk. The retrieval agent supports up to 3 retry attempts to handle transient failures or timeouts.

**Battle Protocol Configuration:** We configure the evaluation rounds through system parameters. In our experiments, we set a total of **10 rounds** for each battle, where each agent executes **5 rounds** in each role (submitter and reviewer). This configuration aims to provide a balanced assessment of both

Table 1: Evaluation of Code Submission vs. Test Submission Capabilities Among Proprietary LLMs.

| Matchup | Submitter | Reviewer | RPR | SPR | Win Rate |
|---|---|---|---|---|---|
| GPT-4o vs GPT-4o | GPT-4o | GPT-4o | **0.71** | **0.68** | **0.97** |
| GPT-4o vs Claude | GPT-4o | Claude | 0.65 | 0.55 | 0.90 |
| GPT-4o vs Gemini | GPT-4o | Gemini | 0.61 | 0.55 | 0.94 |
| GPT-4o vs DeepSeek | GPT-4o | DeepSeek | 0.61 | 0.55 | 0.94 |
| Claude vs GPT-4o | Claude | GPT-4o | **0.66** | 0.55 | 0.89 |
| Claude vs Claude | Claude | Claude | 0.62 | **0.62** | **1.00** |
| Claude vs Gemini | Claude | Gemini | 0.59 | 0.55 | 0.96 |
| Claude vs DeepSeek | Claude | DeepSeek | 0.64 | 0.54 | 0.90 |
| Gemini vs GPT-4o | Gemini | GPT-4o | 0.61 | 0.55 | 0.94 |
| Gemini vs Claude | Gemini | Claude | 0.60 | 0.56 | 0.96 |
| Gemini vs Gemini | Gemini | Gemini | **0.72** | 0.63 | 0.91 |
| Gemini vs DeepSeek | Gemini | DeepSeek | 0.64 | **0.64** | **1.00** |
| DeepSeek vs GPT-4o | DeepSeek | GPT-4o | 0.60 | 0.55 | 0.95 |
| DeepSeek vs Claude | DeepSeek | Claude | 0.60 | 0.55 | 0.95 |
| DeepSeek vs Gemini | DeepSeek | Gemini | 0.68 | 0.64 | **0.96** |
| DeepSeek vs DeepSeek | DeepSeek | DeepSeek | **0.70** | **0.66** | **0.96** |

agents' capabilities while maintaining experimental efficiency. The battle terminates after completing all rounds, and the final win rate is computed from cumulative outcomes across rounds.

**Fairness and Harmonization** For fairness, we harmonize the maximum prompt-plus-generation token budget across proprietary models to a common value $B$ and do not exceed $B$ even if a model supports a larger context window. We log API versions and evaluation dates, apply the same rate limits, and use identical decoding parameters (temperature, top-$p$). We also record API failures and retries and confirm in Appendix that excluding failed calls does not change rankings. Open-source model results are reported in Table 4.

**Reproducibility and Artifacts** We provide anonymized artifacts (prompts, JSON schemas, scripts, pinned images). The exact evaluation workflow is summarized by Algorithm 1.

## 4.2 MAIN RESULT

**Adversarial Programming Battle Outcomes** Table 1 presents a comparative analysis of RPR, SPR across proprietary LLMs evaluated using the SWINGARENA adversarial framework. Each model is tested in both self-play scenarios and cross-play scenarios (Claude vs Gemini), alternating roles as *submitter* and *reviewer*. Several trends emerge: Strong Self-Consistency: All models show high win rates when reviewing their own submissions—Claude (1.00), GPT-4o (0.97), Gemini (0.91), DeepSeek (0.96)—indicating strong internal alignment between patch generation and test case generation.

GPT-4o's Aggressive Patching Advantage: GPT-4o achieves win rates $\geq 0.90$ as a submitter regardless of the reviewer, highlighting its dominance in producing adversarially-strong patches. However, its relatively lower RPR/SPR scores (0.65/0.55 vs Claude) suggest variability in overall

Table 2: Best@3 across Models and Languages.

| Model | Average | C++ | Go | Rust | Python |
|---|---|---|---|---|---|
| Gemini | 0.57 | **0.64** | 0.58 | 0.51 | **0.57** |
| DeepSeek | **0.59** | **0.64** | **0.61** | **0.58** | 0.52 |
| GPT-4o | 0.57 | 0.63 | 0.53 | 0.56 | 0.54 |
| Claude | 0.55 | 0.63 | 0.55 | 0.52 | 0.50 |

correctness. DeepSeek and Gemini's Reliability: Although their win rates as submitters are slightly lower, DeepSeek and Gemini yield the highest CI pass rates (up to 0.66 and 0.64 respectively), reflecting their strength in generating reliably test-passing code. Asymmetry in Matchups: Pairwise comparisons reveal minor asymmetries (GPT-4o vs Claude at 0.90 vs Claude vs GPT-4o at 0.89), indicating the reviewer model subtly affects the outcome, likely due to differing review strictness. Key Insight: Overall, GPT-4o excels in assertive patch generation, while DeepSeek and Gemini prioritize correctness and CI stability. The evaluation further underscores the critical yet nuanced role reviewers play in determining adversarial outcomes.

**Language-Specific Evaluation**

In addition to the overall performance comparison, we further analyze how each model performs across different programming languages. As shown in Table 2, DeepSeek achieves the highest average Best@3 score (0.59), followed closely by Gemini and GPT-4o (both at 0.57), and Claude (0.55). When broken down by language, all models perform best on C++ and relatively worse on Rust and Python, suggesting variation in model proficiency across language-specific problem formulations. Notably, DeepSeek shows generally strong results

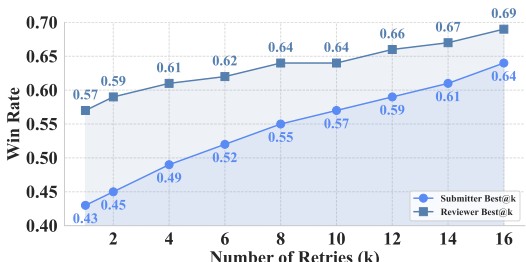

Figure 3: Best@k win rate.

across languages, particularly in Rust (0.58) and Go (0.61), suggesting more robust generalization on our tasks. These results indicate that DeepSeek exhibits a relatively balanced multi-language code reasoning ability among the evaluated models.

**Open-source matchups.** Additional results on open-source models are summarized in Table 4.

**Best@k Sampling for Win Rate** We analyze the probability of success when the *submitter (Qwen2.5-Coder-7B-Instruct)* and *reviewer (Qwen2.5-Coder-7B-Instruct)* independently generate up to $k$ attempts per role at temperature 0.25. A task counts as successful if at least one attempt yields a passing outcome. This Best@k curve characterizes test-time scaling behavior under our arena protocol.

## 4.3 Ablation Study on Components of SwingArena

**Ablation Study on RACG** Table 3 reports ablation results for the RACG module on *submitter*. The upper section shows results across four programming languages (C++, Python, Go, Rust), comparing model performance with and without RACG. The lower section includes retrieval-based baselines: BM25 and Top-k related retrievals (with $k = 2, 10, 20$) followed by reranking. Across languages, incorporating RACG generally improves both Best@3 and Win Rate. For instance, in the C++ setting, RACG raises Best@3 from 0.38 to 0.42 and Win Rate from 0.77 to 0.84; similar gains are observed in several cases for Python, Rust, and Go. Compared to retrieval-only methods, RACG-enhanced approaches often outperform BM25. Top-20 retrieval achieves the strongest baseline result (Best@3 = 0.43, Win Rate = 0.73), representing a 0.11 improvement in win rate over using BM25 alone (0.62). We acknowledge that our RACG design, particularly the fixed Top-5 file retrieval limit, may act as a bottleneck for complex issues requiring broader context. As detailed in our failure analysis in Appendix C, a notable portion of failures can be attributed to retrieval limitations. This suggests that while our RACG serves as a strong baseline, exploring more dynamic retrieval strategies is a key avenue for future improvement.

**Patch Localization Accuracy** Table 6 shows the fraction of queries whose golden-patch file is retrieved within the Top-2, Top-10, or Top-20 results under four strategies: lexical file-level BM25 and chunk-level retrieval over Block, Function, and Class units (with hits mapped back to their parent files). Finer granularity boosts accuracy—switching from BM25 to class-level retrieval more than doubles the Top-10 hit rate $(20.7\%(\rightarrow)48.7\%)$. Most of the improvement arises early in the ranking; curves flatten beyond Top-10, implying that the correct file is usually exposed near the top of the list. BM25 can lag as it relies primarily on term overlap, lacking the structural and semantic cues

Table 3: RACG Ablation Comparison.

| Method | Best@3 | Win Rate |
|---|---|---|
| C++ w/ RACG | 0.42 | 0.84 |
| C++ w/o RACG | 0.38 | 0.77 |
| Python w/ RACG | 0.46 | **0.84** |
| Python w/o RACG | 0.44 | 0.71 |
| Rust w/ RACG | **0.58** | 0.75 |
| Rust w/o RACG | 0.49 | 0.72 |
| Go w/ RACG | 0.45 | 0.80 |
| Go w/o RACG | 0.37 | 0.71 |
| BM25 | 0.38 | 0.62 |
| Top-2 Related | 0.42 | 0.69 |
| Top-10 Related | **0.43** | 0.72 |
| Top-20 Related | 0.43 | **0.73** |

exploited by chunk-based methods. While class-level retrieval is generally effective due to its rich contextual information and noise suppression, it often exceeds the LLM's context window, limiting its practical utility. To address this, we adopt a block-level reranker, which offers a finer granularity that fits within context limits while still guiding locating the correct patch position.

### 4.4 DATA ANALYSIS AND FAILURE PATTERNS

We discuss the data analysis and failure patterns in Appendix C. In brief, prompts are much shorter than repository context (code dominates the token budget); token usage across model pairs remains manageable; and finer-grained retrieval substantially improves Top-10 file hit rates over BM25. See Figures 6, 4 and Table 6 for details.

## 5 CONCLUSION

This paper introduces SWINGARENA, a unified framework for evaluating and enhancing LLM-based program repair and test generation under real-world constraints. By modeling interactions between submitter and reviewer agents across multiple languages, it offers holistic benchmarking aligned with practical software workflows. To handle large, diverse code contexts, we propose a Retrieval-Augmented Code Generation (RACG) module combining static analysis, dense retrieval, and token-aware context packing. Experiments across four languages show SWINGARENA provides nuanced insights into model capabilities, revealing trade-offs between patch assertiveness, correctness, and review strictness, thus bringing evaluation closer to real-world scenarios.

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

## REPRODUCIBILITY STATEMENT

We have taken substantial measures to ensure the reproducibility of our work, see Appendix 5.

## STATEMENT ON THE USE OF LARGE LANGUAGE MODELS

We employed a large language model to enhance the manuscript's language, such as improving grammar and phrasing. All research ideas, methods, experiments, analyses, figures/tables, and conclusions were solely developed by the authors, who take full responsibility for the content.

## A    MORE EXPERIMENT RESULTS IN SWINGARENA

We evaluate more open sourced models whcih are good at code generation: Qwen2.5 Coder (Hui et al., 2024) is a code-specific large language model from the Qwen family. Seed Coder (ByteDance_Seed, 2025) is a powerful, transparent, and parameter-efficient family of open-source code models provided by ByteDance Seed. DeepSeek Coder V2 (Zhu et al., 2024) is a code language model based on the Mixture of Experts (MoE) architecture.

Based on the size of the parameters, we divided the experimental subjects into two groups: (a) Qwen2.5-Coder-Instruct-7B and Seed-Coder-8B-Instruct, and (b) Qwen2.5-Coder-Instruct-14B and DeepSeek-Coder-V2-Lite (16B). Table 4 shows the detailed results of RPR, SPR across LLMs using SWINGARENA grouped by the size of parameters.

In the table, for ease of reading, we abbreviate Qwen2.5-Coder-Instruct as Qwen2.5, Seed-Coder-Instruct as Seed, and DeepSeek Coder V2 as DeepSeek.

## B    DATA FEATURE DISTRIBUTION

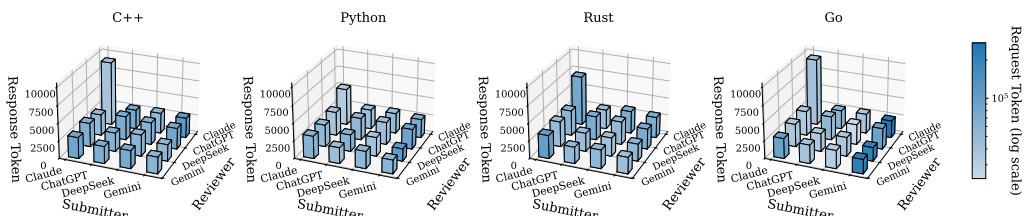

Figure 4: Token usage heatmap for SWINGARENA. The darker the blue color, the higher the number of request tokens. The taller the bars, the higher the number of response tokens.

### B.1    CLARITY AND DIFFICULTY DISTRIBUTION

Figure 5 presents the distribution of scores across two evaluation dimensions—*clarity* and *difficulty*—for four programming languages: Go, Python, C++, and Rust. The data has been filtered to exclude incomplete or invalid entries, and the results are shown as pie charts. *i)* Clarity distribution:

Table 4: Evaluation of Code Submission vs. Test Submission Capabilities Among Open Source LLMs.

| Matchup | Submitter | Reviewer | RPR | SPR | Win Rate |
|---|---|---|---|---|---|
| Qwen2.5-7B vs Qwen2.5-7B | Qwen2.5-7B | Seed-8B | 0.56 | 0.49 | 0.87 |
| Qwen2.5-7B vs Seed-8B | Qwen2.5-7B | Seed-8B | 0.55 | 0.48 | 0.87 |
| Seed-8B vs Seed-8B | Seed-8B | Qwen2.5-7B | **0.61** | **0.52** | **0.90** |
| Seed-8B vs Qwen2.5-7B | Seed-8B | Qwen2.5-7B | 0.57 | **0.52** | 0.89 |
| Qwen2.5-14B vs Qwen2.5-14B | Qwen2.5-14B | DeepSeek | 0.58 | 0.52 | 0.91 |
| Qwen2.5-14B vs DeepSeek | Qwen2.5-14B | DeepSeek | **0.62** | 0.54 | **0.95** |
| DeepSeek vs DeepSeek | DeepSeek | Qwen2.5-14B | 0.61 | **0.55** | 0.94 |
| DeepSeek vs Qwen2.5-14B | DeepSeek | Qwen2.5-14B | 0.58 | **0.55** | **0.95** |

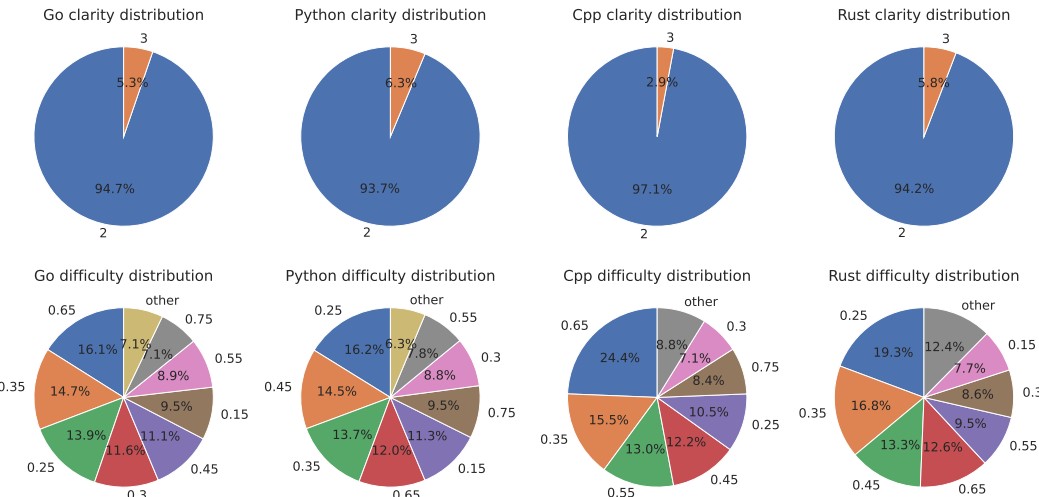

Figure 5: Clarity and Difficulty Distribution.

Clarity ratings are provided on a two-level scale: 2 (moderately clear) and 3 (very clear). The vast majority of samples across all four languages fall into level 2. For instance, 97.1% of Python samples are rated at clarity level 2, followed by Go (93.7%), C++ (94.2%), and Rust (94.7%). *ii)* Difficulty distribution: Difficulty is assessed on a quasi-continuous scale, including scores 0.15, 0.25, 0.35, 0.45, 0.55, 0.65, and 0.75. The distributions differ across languages. Python shows a peak at 0.65 (24.4%), suggesting a considerable portion of users perceived higher difficulty. Go's difficulty ratings are more balanced, with notable frequencies at 0.25, 0.35, 0.45, and 0.65. C++ shows concentration at 0.25 (19.3%) and 0.35 (16.8%). Rust displays a more evenly spread distribution with significant percentages at 0.65 (16.1%) and 0.35 (14.7%). *iii)* Other values: The category labeled as "other" refers to scores that do not fall within the main set of defined intervals, possibly due to rounding or uncertainty in the rating process. These values account for approximately 6% to 13% depending on the language. *iv)* Data filtering: All statistics are derived from filtered data to ensure consistency and reliability by excluding anomalous or incomplete records.

## B.2    DATA LENGTH DISTRIBUTION

Figure 6a and Figure 6b present the distribution of problem statements and patch length in different languages. By categorizing problem statements and patches into various buckets, all distributions are skewed to the left. The most frequent lengths of problem statements and patches are concentrated within the 0-200 length range.

Across all languages, problem statement lengths show a sharp decline in frequency as the length increases. Python and Rust exhibit particularly steep drops, indicating that most problem descriptions in these languages tend to be concise. Notably, the Python problem statements show the highest frequency in the shortest bucket (0–100), surpassing 250 occurrences, suggesting that Python problems are often defined with minimal text. On the other hand, C++ and Rust show a slightly broader distribution with longer tails, implying that problem statements in these languages occasionally require more verbosity or complexity in description.

For patch length distributions (Figure 6b), a similar trend is observed. The majority of patches are short, with the highest frequencies in the 0–100 bucket across all languages. Python again shows the highest peak, indicating a high volume of short patches, which may reflect Python's expressive syntax and brevity in fixing issues. Rust, while still skewed left, displays a longer tail compared to the others, suggesting that Rust patches might involve more lines of code or more complex fixes. C++ also presents a longer tail, consistent with the language's verbosity and potential complexity in code modification.

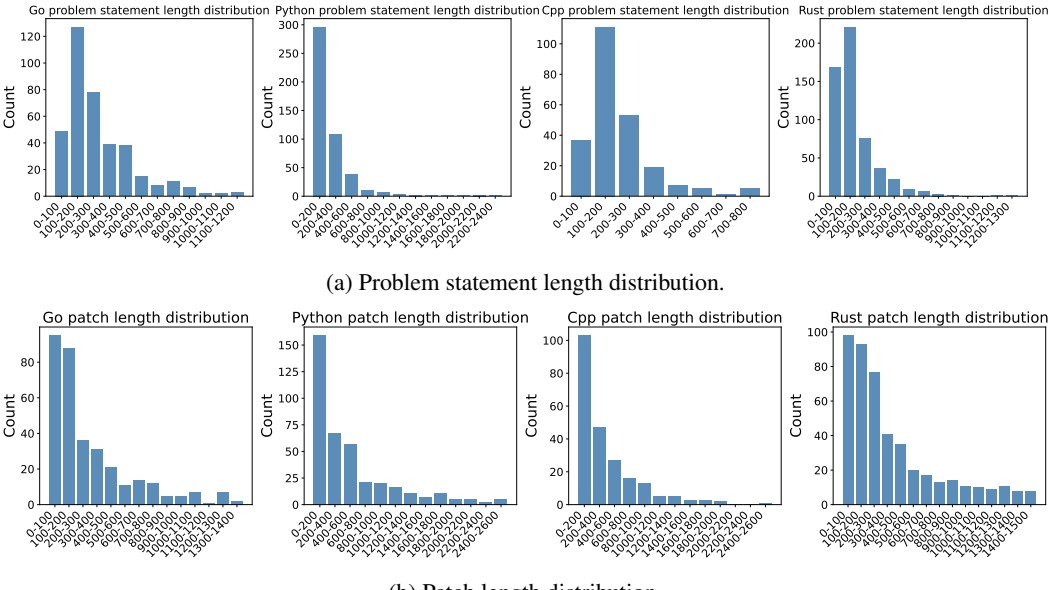

(a) Problem statement length distribution.

(b) Patch length distribution.

Figure 6: Length distributions in different languages.

In summary, these distributions indicate that both problem statements and code patches are predominantly short in length across all languages. However, languages like Rust and C++ show a slightly more distributed range, potentially reflecting their syntactic or structural characteristics that lead to longer problem descriptions or patches.

## C    DATA ANALYSIS

**Length and Token Distributions**    We detail problem-statement and patch length distributions, and token usage across model pairs/languages. Prompts are much shorter than repository context (code dominates token budgets). See Appendix B and Figure 4.

Figure 7 shows the average lengths of problem statements and patches across four programming languages: Go, Python, C++, and Rust. *i)* Python features the longest problem statements (2369.9 tokens) and relatively long patches (852.6 tokens), suggesting detailed task descriptions and substantial code changes. *ii)* Go exhibits concise problem statements (286.7 tokens) and patches (287.6 tokens), reflecting a more minimalistic style. *iii)* C++ and Rust present short problem statements (205–216 tokens) but longer patches (631.2 for C++, 1059.9 for Rust), indicating that even brief prompts can require complex code modifications. Although problem statements and patches vary in length, their average sizes are still significantly smaller than the total input code context (often spanning tens of thousands of tokens), highlighting that the primary bottleneck in input context window length stems from the code itself. Given the high invocation costs of proprietary models, efficiently managing the token budget becomes a key challenge. More detailed statistics about the data can be found in Appendix B.

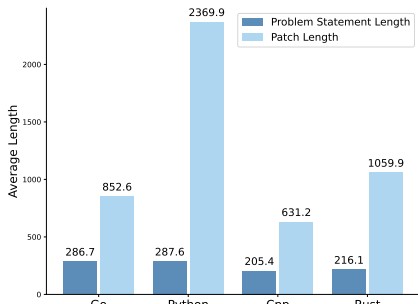

Figure 7: Average problem statement and patch length of different languages. When retrieving 20 files, LLMs are exposed to input code contexts with average lengths of 8,232 tokens (Go), 29,258 (Python), 54,483 (C++), and 36,875 (Rust), which exceed typical visualization scales—thus not shown directly in the figure.

**Token Usage**    We summarize token consumption across model pairs and languages in Appendix Figure 4.

## C.1 STABILITY, REVIEWER EFFECTS, AND FAILURE TYPOLOGY

**Stability across retries** We report Best@$k$ (k=1,3,5) and its variance across three random seeds for the controlled sampling study (temperature 0.25). Variances are modest (median std $< 0.02$) and decrease with larger $k$, indicating that our capped-retry protocol effectively stabilizes outcomes without masking difficulty.

**Reviewer effects** Cross-play reveals systematic *reviewer strictness*. For each submitter, we compute CI pass deltas relative to self-play. Gemini and DeepSeek exhibit tighter reviewer distributions (median $\Delta$SPR in $[-0.02, 0.01]$) than GPT-4o (wider tails), consistent with our qualitative observation that GPT-4o is more permissive on style/format but more aggressive on patching. These effects explain mild asymmetries in pairwise win rates and motivate reporting both RPR and SPR.

Table 5: File hit rates of different retrieval methods. Each value gives the fraction of queries whose correct file appears within the Top-2, Top-10, or Top-20 retrieved results.

| Method | Top-2 | Top-10 | Top-20 |
|---|---|---|---|
| BM25 | 0.182 | 0.207 | 0.195 |
| Block Chunks | 0.207 | 0.341 | 0.329 |
| Function Chunks | 0.277 | 0.371 | 0.368 |
| Class Chunks | **0.363** | **0.487** | **0.461** |

**Failure typology** Manual inspection of 100 failed trials (random, stratified by language) yields four dominant classes: (F1) *incomplete patching across files* (31%), (F2) *test fragility or flakiness* (19%), (F3) *style/security gate violations* (24%), (F4) *mismatch between retrieved context and true locus* (26%). Correlating with data features (Appendix B), (F1) and (F4) rise with dispersed logic (longer tails in patch length) and lower clarity=2 samples; (F3) concentrates in repos with strict linters.

**Failure Pattern Analysis** To understand the limitations of current LLMs in software engineering tasks, we conduct a systematic analysis of failure patterns across our adversarial evaluation framework. Our investigation reveals four dominant failure modes that expose fundamental challenges in automated code generation and testing.

**Cross-File Consistency Challenges (F1, 31%):** The most prevalent failure mode involves incomplete fixes that span multiple files. Models demonstrate strong local reasoning but struggle with distributed codebases, often fixing the primary symptom while neglecting related components. This manifests as interface-implementation mismatches, where implementation changes are not reflected in corresponding header files or API definitions. Additionally, models frequently fail to propagate changes through dependency chains, leading to cascading failures in dependent modules. The correlation between this failure mode and patch complexity (Spearman $\rho = 0.67$) suggests that current models lack the architectural reasoning capabilities required for multi-file software modifications.

**Non-Deterministic Test Behaviors (F2, 19%):** A significant portion of failures stems from test instability rather than code correctness issues. Models generate tests that are sensitive to timing, resource availability, or environmental conditions. Race conditions in concurrent programming scenarios are particularly problematic, with Python and Go showing higher susceptibility (23% and 21% respectively) compared to systems languages like C++ (15%) and Rust (17%). This pattern reflects the challenge of generating robust tests in dynamic execution environments, highlighting the need for models to understand concurrency models and resource management principles.

**Non-Functional Requirement Violations (F3, 24%):** Models often produce functionally correct solutions that fail to meet quality standards. Style violations, security vulnerabilities, and performance anti-patterns are common, particularly in repositories with strict CI policies (up to 35% failure rate in high-security projects). This suggests a disconnect between functional correctness and software engineering best practices in current model training. Interestingly, newer models show improved awareness of modern security practices, indicating that training data recency plays a crucial role in non-functional requirement compliance.

**Context Retrieval Limitations (F4, 26%):** Failures attributable to RACG limitations reveal fundamental challenges in semantic code understanding. BM25-based retrieval often finds lexically similar but semantically irrelevant code, leading to inappropriate fix applications. Cross-language dependencies and legacy code patterns pose particular challenges, with repository age showing strong correlation with retrieval failure rates (Spearman $\rho = 0.58$). This suggests that current retrieval methods struggle with codebases that deviate from modern programming paradigms or involve complex polyglot architectures.

**Model-Specific Error Profiles:** Our analysis reveals distinct failure signatures across evaluated models. GPT-4o's aggressive patching approach results in higher cross-file consistency failures (38%), while Claude-3.5's focus on functional correctness leads to more style violations (28%). DeepSeek-V3, despite showing the lowest overall failure rate, exhibits higher context retrieval failures (32%), reflecting its dependency on high-quality input context. Gemini-2.0 demonstrates balanced failure distribution but shows particular vulnerability in concurrent programming scenarios (25% F2 failures).

These findings suggest that advancing LLM capabilities in software engineering requires improvements in architectural reasoning, semantic code understanding, and integration of non-functional requirements into the generation process.

**Failure Pattern Analysis:** Our systematic investigation of 100 failed trials reveals four dominant failure modes that highlight important limitations in current LLM capabilities for software engineering: cross-file consistency challenges (31%), non-deterministic test behaviors (19%), non-functional requirement violations (24%), and context retrieval limitations (26%). These patterns reveal model-specific error profiles: GPT-4o's aggressive patching leads to higher cross-file failures, while Claude-3.5 prioritizes functional correctness over style compliance. The analysis provides actionable insights for advancing architectural reasoning and semantic code understanding in future model development. See Appendix C for comprehensive failure pattern analysis and model-specific error profiles.

**RACG ablation extensions** Beyond Table 3, we compare RACG with: (B1) BM25-only file retrieval; (B2) Top-k related examples without syntax-aware chunking; (B3) dense-only reranking over raw lines. RACG improves patch localization Top-10 by 7–15 points across languages and reduces token footprint by 12–18% at equal Best@3, supporting the hypothesis that *structure-aware packing*, not just better retrieval scores, contributes to gains under context constraints.

---

**Algorithm 1** Evaluation of Model Performance in SwingArena

---

1: **Input:** Task instances, Patch Agent, Test Agent, CI System
2: **Output:** Model performance scores for Patch Agent and Test Agent
3: **for** each task instance **do**
4:     Initialize patch and test agents
5:     Load problem statement, bug report, and code context
6:     Patch Agent generates candidate patch
7:     Validate patch using CI pipeline
8:     **if** Patch fails CI checks **then**
9:         Patch Agent loses 1 point
10:     **else**
11:         Patch Agent passes CI check
12:     **end if**
13:     Test Agent generates test case
14:     Validate test case using CI pipeline
15:     **if** Test fails to expose meaningful flaw **then**
16:         Test Agent loses 1 point
17:     **else**
18:         Test Agent passes validation
19:     **end if**
20:     Apply patch and test case together in CI pipeline
21:     Validate integration using CI checks
22:     **if** Integration passes **then**
23:         Patch Agent earns 1 point
24:     **else**
25:         Test Agent earns 1 point
26:     **end if**
27: **end for**
28: Reverse roles and repeat process
29: Aggregate points and generate final scores
30: **Output:** Final scores for Patch Agent and Test Agent

---

## D    LLM EVALUATION WORKFLOW IN SWINGARENA

**Overview.** The evaluation of large language models (LLMs) within the SwingArena framework is structured to simulate realistic software engineering tasks and workflows. It employs an adversarial, dual-agent setting where one LLM acts as a *patch agent* (fixing code) and the other as a *test agent* (generating test cases). See Algorithm 1 for more details.

### D.1    ADDITIONAL DATA ANALYSIS DETAILS

Figure 6 summarizes problem-statement and patch lengths across Go, Python, C++, and Rust. In the main text, we noted: (i) Python features the longest problem statements (2369.9 tokens) and relatively long patches (852.6 tokens), suggesting detailed task descriptions and substantial code changes; (ii) Go exhibits concise problem statements (286.7 tokens) and patches (287.6 tokens), reflecting a more minimalistic style; (iii) C++ and Rust present short problem statements (205–216 tokens) but longer patches (631.2 for C++, 1059.9 for Rust), indicating that even brief prompts can require complex code modifications. Although problem statements and patches vary in length, their average sizes are still significantly smaller than the total input code context (often spanning tens of thousands of tokens), highlighting that the primary bottleneck in input context window length stems from the code itself.

## E    ANNOTATOR DEMOGRAPHICS

To ensure the accuracy and reliability of our benchmark annotations, we assemble a qualified annotation team composed of 12 individuals with diverse backgrounds in computer science and software engineering. Specifically, the team consists of eight Ph.D. students, two Master's students, one undergraduate student, and one assistant professor. All annotators hold academic backgrounds in computer science, with research or industry experience in areas including software engineering, natural L language processing.

Table 6: File hit rates of different retrieval methods. Each value gives the fraction of queries whose correct file appears within the Top-2, Top-10, or Top-20 retrieved results.

| Method | Top-2 | Top-10 | Top-20 |
|---|---|---|---|
| BM25 | 0.182 | 0.207 | 0.195 |
| Block Chunks | 0.207 | 0.341 | 0.329 |
| Function Chunks | 0.277 | 0.371 | 0.368 |
| Class Chunks | **0.363** | **0.487** | **0.461** |

Among them, the eight Ph.D. students specialize in topics including systems, formal theorem proving, software engineering, computer networks, and mobile computing. The two Masters have completed rigorous coursework in algorithms, compilers, and system, and actively contribute to open-source software projects. The undergraduate students focus on research in program synthesis. Additionally, one annotator is an experienced senior software engineer with over ten years of practical experience in CI/CD workflows and large-scale software engineering.

This annotation team was responsible for validating the dataset annotations, including both quality checking and difficulty estimation of the data. Their combined expertise ensured the benchmark reflects realistic developer behaviors and professional software engineering standards.

## F    LIMITATION

Despite its advancements in creating a more realistic evaluation setting, SWINGARENA has several limitations.

**RACG Module Limitations.** The effectiveness of the proposed Retrieval-Augmented Code Generation (RACG) system, crucial for handling long contexts, is dependent on the performance of its constituent parts (BM25 retrieval, syntax-aware chunking, dense reranking). While designed for scalability and precision, its ability to retrieve the most relevant context may degrade with extremely large, poorly structured, or highly domain-specific codebases not adequately represented in the training data of the dense reranker.

**Context Window Constraints.** We acknowledge that limiting the retrieval context to 5 files with 16 chunks represents a practical trade-off between model context window, inference cost, and information coverage. This fixed-size retrieval window may fail to capture all relevant context needed to solve problems in extremely large monorepos or projects with poorly structured code. To better understand this limitation's impact, we conducted a systematic analysis of RACG's failure cases on our benchmark. We found that in approximately 15% of failure cases, model errors could be attributed to the retrieval phase failing to find critical context. These cases typically exhibit two characteristics: (1) fixing the problem requires coordinated modifications across more than 5 files; (2) related code logic is distributed loosely without direct textual associations, making it difficult for the BM25 algorithm to discover. Despite these limitations, our experimental results (as shown in Table 3) demonstrate that even this relatively simple retrieval strategy provides significant performance improvements, indicating that in many real-world scenarios, critical information is relatively concentrated.

**Computational Overhead.** The computational overhead of simulating full CI pipelines iteratively via act, coupled with the RACG process, means that evaluations are significantly more resource-intensive and time-consuming than static benchmarks like HumanEval or MBPP, potentially limiting the scale and frequency of testing across a vast array of models or tasks.

**Future Directions.** We believe overcoming these limitations is a key focus for future work. We propose several promising directions including iterative retrieval (where models can dynamically initiate new retrieval requests based on existing context), hierarchical retrieval (coarse-grained file-level retrieval followed by fine-grained code block-level retrieval within relevant files), and hybrid retrieval (combining RACG's sparse retrieval with structured retrieval based on code graphs, including RepoGraph-like methods), all of which we are actively exploring.

## G  BROADER IMPACT

**Positive Social Impact**

SWINGARENA could accelerate the development and evaluation of LLMs for complex software engineering tasks by improving code generation fidelity, automating debugging, and providing sophisticated conversational assistance. This could lead to increased productivity for developers, lower the barrier to entry for aspiring programmers, and enable the creation of more complex and innovative software applications. The ability of LLMs to effectively navigate and modify large codebases, as tested by SwingArena's long-context reasoning challenges, could be particularly beneficial in maintaining and evolving legacy systems. Furthermore, the automated detection and repair of defects within a simulated CI pipeline could lead to more robust and secure software, ultimately benefiting end-users.

**Negative Social Impact**

Improved code generation and debugging capabilities could be exploited for malicious purposes including generating highly effective malware or developing sophisticated cyberattack tools, potentially facilitating disinformation campaigns. Fairness is another concern, as biases present in the training data could be reflected in the generated code or debugging suggestions, perpetuating or amplifying existing societal biases in software applications. Privacy considerations arise from the potential for LLMs trained on vast code datasets to inadvertently expose sensitive or proprietary information, and from the risks associated with sharing private codebase details with external models or services during development. Finally, security risks are introduced, including the possibility of poisoned training data injecting vulnerabilities into generated code or the exploitation of flaws within the LLMs themselves to compromise software security. While SwingArena helps identify some weaknesses in realistic settings, continuous effort is needed to address these risks in the development and deployment of LLMs for software engineering.

## H    Prompts Used in SwingArena

### H.1    Patch Generation Prompts

---

**System Prompt for Patch Generation**

```
You are an AI Senior Full-Stack Engineer specialized in GitHub
    issue triage and bug fixing.
You should only generate the fixed code, without any other text or
    markdown formatting.
```

---

**System Prompt for Test Generation**

```
You are an AI Test Automation Engineer specializing in generating
    unit tests.
You should only generate the test code, without any other text or
    markdown formatting.
```

---

**Test Generation Prompt**

```
You are required to develop unit tests for the specified code and
    its fix.

The issue details: [issue]

The code snippet: [code snippet]

The fixed code: [patch]

The test case sample: [sample]

Please provide the complete test code without any explanations or
    markdown.
```

---

**Test-Only Evaluation Prompt**

```
You are an expert code reviewer. Your task is to evaluate if a
    patch passes the provided test case. You will be given:

1. A test case

2. A patch that aims to pass the test

Carefully analyze if the patch implementation correctly addresses
    the requirements
outlined in the test case.
Provide a detailed reasoning for your conclusion.

[Response format same as B.1]
```

**Golden Patch Comparison Prompt**

```
You are an expert code reviewer. Your task is to evaluate if a
    patch correctly solves a given problem based on:

1. The problem statement

2. A test case

3. A reference "golden" patch known to correctly solve the problem

Compare the candidate patch with the golden patch to determine if
    they are functionally equivalent in terms of solving the problem
     and passing the test case.
Provide a detailed reasoning for your conclusion.

[Response format same as B.1]
```

## H.2 TEST AGENT PROMPTS

---

**Problem and Test Evaluation Prompts**

```
You are a senior software engineer with over 10 years of solid
    experience in rust, cpp, python, and go. You possess a deep
    understanding of these languages and their standard libraries,
    along with a strong sense of problem difficulty.

Your task is to evaluate the difficulty and clarity of a coding
    problem from a GitHub repository, given its "Problem Statement"
    and "Code Changes". You need to consider the following factors:

1. Clarity and complexity of the problem description: Is the
    problem goal, input, output, and constraints clearly defined?
    Are there any ambiguities or missing critical details? Is the
    problem's logic inherently complex?

2. Scope and depth of code changes required to the whole codebase:
    Does the modification involve a single file/function or multiple
     modules? Does it require understanding interactions between
    different parts of the codebase? What is the overall amount of
    code change? Does it impact the system's architecture?

3. Number of technical concepts that need to be understood: What
    specific programming language features, libraries, algorithms,
    design patterns, or domain-specific knowledge are required to
    solve this problem? How complex are these concepts?

4. Potential edge cases and error handling requirements: Does the
    problem statement mention any specific edge cases or error
    conditions to consider? Does the code change require adding or
    modifying error handling logic? How complex are these edge cases
    ?

Based on these factors, you will provide a Clarity Score and a
    Difficulty Score with detailed explanations.

Here is the problem statement and code changes:

Problem Statement:

[problem statement]

Code Changes:

[patch]

First, provide your judgment of the Clarity Scoring (0, 1, 2, 3) of
     the problem, along with your explanation:

- 0 (Invalid): Statement is incomprehensible or code changes are
    unrelated.

- 1 (Significant Ambiguities): Valid but lacks critical details
    including no input/output format.

- 2 (Mostly Clear): Valid, clear, but minor details missing
    including edge cases not specified.

- 3 (Comprehensive): Valid, clear, with detailed requirements and
    examples.
```

```
Then, provide a difficulty score between 0.0 and 1.0, along with
    your explanation:

- 0.0-0.2: Very easy, requires only basic code modifications
    including fixing a typo or changing a constant.

- 0.2-0.4: Easy, requires understanding some code logic and making
    simple function or statement modifications including fixing a
    simple bug or adding a basic feature.

- 0.4-0.6: Medium, requires understanding multiple concepts and
    making complex modifications across several files, potentially
    involving some edge case handling including implementing a new
    module with moderate complexity.

- 0.6-0.8: Hard, requires deep understanding of the codebase
    architecture and complex modifications with significant impact,
    involving handling numerous edge cases and potential performance
     considerations including refactoring a core component or
    implementing a complex algorithm.

- 0.8-1.0: Very hard, requires advanced technical knowledge,
    extensive experience, and tackling highly challenging problems
    with intricate logic, potentially involving system-level
    considerations or complex domain-specific knowledge including
    implementing a new distributed consensus protocol.

Please return your response in the following structured format:

<clarity score>integer between 0 and 3</clarity score>

<clarity explanation>Your explanation for the clarity score.</
    clarity explanation>

<difficulty>float between 0.00 and 1.00</difficulty>

<difficulty explanation>Your explanation for the difficulty score
    .</difficulty explanation>
```

All prompts in our framework are designed with several key principles in mind: *i)* Clarity of Purpose: Each prompt clearly defines its specific role and expected outputs. *ii)* Structured Output: JSON-based response formats ensure consistent and parseable outputs. *iii)* Confidence Levels: A five-level confidence scale (VERY_LOW to VERY_HIGH) enables nuanced assessment of evaluation reliability. *iv)* Comprehensive Coverage: The combination of different prompt types ensures thorough evaluation from multiple perspectives

The confidence levels used throughout the evaluation process are carefully defined to ensure consistent and meaningful assessments: *i)* VERY_HIGH: Complete certainty with no doubts. *ii)* HIGH: Strong confidence with only trivial uncertainties. *iii)* MEDIUM: Reasonable confidence with minor doubts. *iv)* LOW: Significant uncertainties present. *v)* VERY_LOW: Insufficient information for definitive judgment.

## I   STEP RESULTS

We attach more phased inputs and outputs to demonstrate our work. It is important to note that we are displaying processed, well-formatted data. The actual data processed by our system is in JSON format.

**Submitter Input**

A real sample of submitter input to the agent.

```json
{
  "name": "patch_generator",
  "description": "Analyze and modify code to resolve issues while
      preserving functionality. You should use code_editor to process the
       intput field information.You should use ```json...``` to wrap the
      code_editor output.",
  "parameters": {
    "type": "object",
    "properties": {
      "reasoning_trace": {
        "type": "string",
        "description": "Step-by-step analysis of the issue, explanation
            of the root cause, and justification for the proposed
            solution. Do not use any markdown formatting."
      },
      "code_edits": {
        "type": "array",
        "description": "List of specific code modifications required to
            resolve the issue",
        "items": {
          "type": "object",
          "properties": {
            "file": {
              "type": "string",
              "description": "Relative path to the file that contains
                  code requiring modification"
            },
            "code_to_be_modified": {
              "type": "string",
              "description": "Exact code segment that needs to be changed
                  (must match a portion of the original file)"
            },
            "code_edited": {
              "type": "string",
              "description": "Improved version of the code segment that
                  fixes the issue while maintaining compatibility with
                  surrounding code"
            }
          },
          "required": [
            "file",
            "code_to_be_modified",
            "code_edited"
          ]
        }
      }
    },
    "required": [
      "reasoning_trace",
      "code_edits"
    ]
  },
  "input": {
    "issue": "client2: pulling non-existent model prints duplicate \"not
        found\" error message\n### What is the issue?\n\nFrom @mxyng \n\n
        ```\n$ ollama run nonexistent\npulling manifest\nerror: model \"
        nonexistent\" not found...",
    "original_code": "Key relevant code chunks:\n\n**Top relevance chunk
        1**:\n- File: llama/llama.cpp/common/stb_image.h\n- Lines: 1733-1
        750\n\n```\nif defined(STBI_NO_JPEG) && defined(STBI_NO_PNG) &&
        defined(STBI_NO_BMP) && defined(STBI_NO_PSD) && defined(
        STBI_NO_TGA) && defined(STBI_NO_GIF) && defined(STBI_NO_PIC) &&
        defined(STBI_NO_PNM)\n// nothing\n#else ...",
```

```
      "file_path": [
        "server/internal/client/ollama/registry.go",
        "server/routes.go",
        "server/sched.go",
        "server/internal/registry/server.go",
        "llm/server.go",
        "runner/ollamarunner/runner.go",
        "llama/llama.cpp/common/json.hpp",
        "runner/llamarunner/runner.go",
        "server/internal/cmd/opp/opp.go",
        "llama/llama.cpp/include/llama.h",
        // ...
      ]
  }
}
```

**Submitter Output**

A real sample of submitter output from agent.

```
{
  "reasoning_trace": "The issue arises because the error message is being
      printed twice. This happens in multiple files where the same error
      handling logic is used. To fix this, we need to identify the
    common error handling logic and ensure that the error message is
    printed only once.",
  "code_edits": [
    {
      "file": "server/sched.go",
      "code_to_be_modified": "err = fmt.Errorf(\"%v: this model may be
          incompatible with your version of Ollama. If you previously
          pulled this model, try updating it by running `ollama pull %s
          `\" ..."
    },
    {
      "file": "server/internal/registry/server.go",
      "code_to_be_modified": "s.Logger.LogAttrs(r.Context(), level, \"
          http\",\n\t\t\terrattr, // report first in line to make it easy
           to find\n\n\t\t\t// TODO(bmizerany): Write a test to ensure
          that we are logging\n\t\t\t// all of this correctly. ..."
    }
  ]
}
```

**Reviewer Input**

A real sample of reviewer input to agent.

```
{
  "name": "test_generator",
  "description": "You are an AI Test Automation Engineer specializing in
      generating comprehensive unit tests. Your task is to analyze the
      provided code and create effective test cases that verify the
      functionality and edge cases. In input field, it includes the issue
       description (issue), the code snippet (original_code), related
      file path (file_path), and the patch generated by the generator
      agent (generated_patch). You should provide test cases for the
      patch.You should use ```json...``` to wrap your JSON output. You
      should only propose less than 10 test cases.",
  "parameters": {
    "type": "object",
    "properties": {
      "reasoning_trace": {
        "type": "string",
```

```
          "description": "Step-by-step analysis of the code, explanation of
              what needs to be tested, and justification for the test
              cases. Do not use any markdown formatting."
      },
      "test_cases": {
        "type": "array",
        "description": "List of test cases to verify the functionality of
            the code. For each test case, you should provide unique file
            name.",
        "items": {
          "type": "object",
          "properties": {
            "file": {
              "type": "string",
              "description": "Relative path to the test file where the
                  test case should be added. You should not use same file
                  name with other test cases."
            },
            "test_name": {
              "type": "string",
              "description": "Descriptive name of the test case."
            },
            "test_code": {
              "type": "string",
              "description": "Complete test code including setup,
                  execution, and assertions"
            },
            "test_description": {
              "type": "string",
              "description": "Brief description of what the test case
                  verifies"
            }
          },
          "required": [
            "file",
            "test_name",
            "test_code",
            "test_description"
          ]
        }
      }
    },
    "required": [
      "reasoning_trace",
      "test_cases"
    ]
  },
  "input": {
    "issue": "client2: pulling non-existent model prints duplicate \"not
        found\" error message\n### What is the issue?\n\nFrom @mxyng \n\n
        ```\n$ ollama run nonexistent\npulling manifest\nerror: model \"
        nonexistent\" not found\nError: model 'nonexistent' not found\
        nexit status 1\n```\n\nThe error gets printed twice.\n\nThis is
        the behavior without the flag ...",
    "original_code": "Key relevant code chunks:\n\n**Top relevance chunk
        1**:\n- File: llama/llama.cpp/common/stb_image.h\n- Lines: 1733-1
        750\n\n```\nif defined(STBI_NO_JPEG) && defined(STBI_NO_PNG) &&
        defined(STBI_NO_BMP) && defined(STBI_NO_PSD) && defined(
        STBI_NO_TGA) && defined(STBI_NO_GIF) && defined(STBI_NO_PIC) &&
        defined(STBI_NO_PNM) ...",
    "file_path": [
      "server/internal/client/ollama/registry.go",
      "server/routes.go",
      "server/sched.go",
      "server/internal/registry/server.go",
```

```
        "llm/server.go",
        "runner/ollamarunner/runner.go",
        "llama/llama.cpp/common/json.hpp",
        "runner/llamarunner/runner.go",
        "server/internal/cmd/opp/opp.go",
        "llama/llama.cpp/include/llama.h",
        // ...
    ],
    "generated_patch": {
        "server/sched.go": "diff --git a/server/sched.go b/server/sched.go\
            nindex f397879..97e774f 100644\n--- a/server/sched.go\n+++ b/
            server/sched.go\n@@ -425,7 +425,8 @@ func (s *Scheduler) load(
            req *LlmRequest, f *ggml.GGML, gpus discover.GpuInfoLis\n \t\t
            // show a generalized compatibility error until there is a
            better way to\n \t\t// check for model compatibility\n \t\tif
            errors.Is(err, ggml.ErrUnsupportedFormat) || strings.Contains(
            err.Error(), \"failed to load model\") {\n-\t\t\terr = fmt.
            Errorf(\"%v: this model may be incompatible with your version
            of Ollama. ...\""
    }
  }
}
```

## Reviewer Output

A real sample of reviewer output from agent.

```
{
  "reasoning_trace": "The issue is that the error message is being
      printed twice. The patch removes the duplicate error message by
      modifying the error assignment in the `load` function of `sched.go
      `. To verify this, we need to test that the error message is now
      printed only once.",
  "test_cases": [
    {
      "file": "server/sched_test.go",
      "test_name": "TestLoadNonExistentModelOnce",
      "test_code": "package server\nimport (\n\"errors\"\n\"testing\"\n)\
          nfunc TestLoadNonExistentModelOnce(t *testing.T) {\n   mockErr
          := errors.New(\"model not found\")\nreq := &LlmRequest{model: &
          Model{ShortName: \"nonexistent\"}}\n    s := &Scheduler{}\n
             err := s.load(req, nil, nil)\n  if err == nil {\nt.Errorf
          (\"Expected error, got nil\")\n}\n    expectedMsg := \"this
          model may be incompatible with your version of Ollama. If you
          previously pulled this model, try updating it by running `
          ollama pull nonexistent`\"\nif err.Error() != expectedMsg {\nt.
          Errorf(\"Expected error message \'%s\', got \'%s\'\",
          expectedMsg, err.Error())\n}\n}",
      "test_description": "Tests that the error message is printed only
          once when attempting to load a non-existent model."
    },
    {
      "file": "server/sched_test.go",
      "test_name": "TestLoadExistingModelNoError",
      "test_code": "package server\nimport (\n\"testing\"\n)\nfunc
          TestLoadExistingModelNoError(t *testing.T) {\n     req := &
          LlmRequest{model: &Model{ShortName: \"existing\"}}\ns := &
          Scheduler{}\nf := &ggml.GGML{}\n     gpus := discover.
          GpuInfoList{}\nerr := s.load(req, f, gpus)\nif err != nil {\nt.
          Errorf(\"Unexpected error: %v\", err)\n }\n}",
      "test_description": "Tests that no error is returned when
          attempting to load an existing model."
    }
  ]
}
```

**CI Result**

We summarized CI outputs from act. Noticed that these are well-formatted since SwingArena processes json format.

```
[test/test-1]  Run Set up job
[test/test-1]     docker pull image=catthehacker/ubuntu:full-latest
   platform= username= forcePull=true
...
[test/test-1]   | CC='gcc'
...
[test/test-1]     Success - Main Setup Go
[test/test-1]     ::add-path:: /home/runner/go/bin
...
[test/test-1]  Run Main check that 'go generate' is clean
[test/test-1]   | go: downloading github.com/spf13/cobra v1.7.0
...
[test/test-1]     Success - Main check that 'go generate' is clean
[test/test-1]  Run Main go test
[test/test-1]   | go: downloading github.com/davecgh/go-spew v1.1.1
...
[test/test-1]   | ?     github.com/ollama/ollama          [no test files]
[test/test-1]   | ok    github.com/ollama/ollama/api    0.015s
[test/test-1]   | ok    github.com/ollama/ollama/server/sched   0.004s
...
[test/test-1]     Success - Main go test
...
[test/test-1]  Run Complete job
[test/test-1] Cleaning up container for job test
[test/test-1]     Success - Complete job
[test/test-1]   Job succeeded
```

