# OpenReview forum: "SWINGARENA: Adversarial Programming Arena for Long-context GitHub Issue Solving"
_ICLR.cc/2026/Conference — ICLR 2026 Oral_

### Official Review · Reviewer_hYHi · 2025-10-28

**Soundness:** 2
**Presentation:** 3
**Contribution:** 3
**Rating:** 6
**Confidence:** 3

**Summary:**

The paper introduces SWINGARENA, a CI-faithful, adversarial code-evaluation arena where an LLM “Submitter” proposes patches and an LLM (or human) “Reviewer” writes tests to break them; the roles can switch across rounds. It ships (i) a rigorously curated multi-language dataset with runnable CI, (ii) a Retrieval-Augmented Code Generation (RACG) pipeline to find relevant files and synthesize fixes, and (iii) arena metrics (e.g., Win Rate, SPR/RPR) to quantify progress. Experiments on real repositories show the arena surfaces harder, more realistic failure modes than static benchmarks and that better retrieval/testing policies materially improve pass rates.

**Strengths:**

1. The adversarial submitter-reviewer paradigm with role-switching is creative and mirrors real-world collaboration better than static benchmarks.
2. Rigorous data curation: Three-stage filtering with human expert validation is commendable.
3. Well-oraganized writing, easy to understand.

**Weaknesses:**

1) **Limited novelty / “integration over invention.”**
   The work skillfully engineers an end-to-end pipeline for realistic SE evaluation, but core techniques (retrieval, ranking, CI emulation, multi-turn prompting) largely reuse known components.
2) **Under-granular ablations for RACG.**
   Current ablations are mostly on/off toggles or coarse retrieval settings, leaving it unclear which sub-module drives gains.
   - **Actionable:** Provide per-component ablations on (i) chunk size & overlap, (ii) reranker family (bi-encoder vs. cross-encoder), (iii) proximity/structure priors (same-dir, same-package), (iv) adaptive Top-k vs. fixed, (v) failure-triggered expansion rules; add mediation analysis to quantify each component’s indirect effect on final win rate.
3) **Language & ecosystem coverage is narrow.**
   Results cover C++/Python/Rust/Go but omit **Java/JS/TS** and build systems (**Maven/Gradle**, **pnpm/yarn/monorepo**). This limits external validity for common enterprise stacks.

**Questions:**

1) **Causal attribution:** How do you isolate the gain from the *adversarial, role-switching* protocol itself (vs. model scale, retrieval strength, prompt length)? Can you run controlled A/Bs holding RACG constant while removing role-switching, and report mediation analysis?

2) **Real-world parity:** How closely do local CI runs match upstream (e.g., GitHub Actions) on pass/fail and runtime? Please provide a paired evaluation with agreement statistics and discuss observed drift.

3) **Gaming & robustness:** Are there “score-hacking” strategies (e.g., submitter creates trivially detectable but non-critical faults; reviewer overfits to diffs)? What constraints or equilibrium analyses prevent metric inflation?

4) **Scaling laws:** As model size, number of rounds, and repo size (files, dependency depth) grow, do we observe consistent scaling behavior? Which has higher marginal return: **more rounds** or **stronger/adaptive retrieval**?

5) **Granular failure taxonomy:** What are the dominant failure modes per language (retrieval miss, build failure, semantic error)? Please add an error decomposition table linking failure modes to specific RACG sub-modules.

6) **Cross-ecosystem generalization:** What minimal changes are needed to support Java (Maven/Gradle) and JS/TS monorepos? Any zero-/few-shot transfer experiments showing the framework’s portability?

7) **Contamination audits:** How did you detect/prevent training–evaluation overlap for both LLMs and the reranker?

8) **Metric resolution:** Beyond overall win rate, can you report **defect-type × difficulty** strata (and CIs) to demonstrate the framework discriminates hard cases rather than being dominated by easy issues?

9) **Adaptive retrieval policy:** Can RACG expand beyond fixed Top-k after early failures? Please report the success-latency trade-off and whether adaptive policies change conclusions.

10) **Cost & reproducibility:** What is the typical wall-clock time and token usage per match under your default settings? Could you provide scripts and a “minimal slice” that reproduces headline results on modest compute?

> If the above analyses (especially #1, #2, #4, #7) are addressed, I would likely raise my overall score.

---

> ### Author Response · Authors · 2025-11-18
> **Response to Reviewer hYHi (1)**
>
> We are very grateful to the reviewers for their detailed review and valuable suggestions.
>
> **Q1: Limited novelty / “integration over invention.”** The work skillfully engineers an end-to-end pipeline for realistic SE evaluation, but core techniques (retrieval, ranking, CI emulation, multi-turn prompting) largely reuse known components.
>
> We appreciate the reviewer’s concern about novelty and the observation that our system leverages existing building blocks such as retrieval, ranking, CI execution, and multi-turn prompting. Our goal, however, is not to propose a new retrieval or modeling algorithm, but to introduce a new evaluation protocol and benchmark that captures CI-driven, adversarial software engineering workflows which are not covered by prior work.
>
> Concretely, our contributions go beyond a straightforward integration of known components:
>
> Adversarial CI arena with dual roles and role switching. Prior benchmarks such as SWE-Bench and its variants focus on single-agent, mostly one-shot patching against static tests. In contrast, SwingArena formalizes a two-agent, adversarial protocol (submitter vs reviewer) over full CI pipelines, with role switching and clearly defined win/lose payoffs. This design enables us to study interactions between patch generation and test generation, which existing static benchmarks cannot capture.
>
> CI-grounded, multi-language dataset and metrics. We curate 2,300 real GitHub issues with solutions across four languages and execute them under repository-native CI workflows (style, coverage, security checks), rather than a single unit-test harness. The accompanying metrics (Win Rate, SPR/RPR, Best@k) are tailored to this adversarial CI setting and allow us to disentangle “aggressive” vs “conservative” patching behaviours.
>
> RACG as a standardized, cross-model baseline for long-context access. We explicitly position RACG not as an algorithmic contribution but as a necessary, multi-language retrieval baseline that harmonizes context budgets across models and languages, controls retrieval variance, and makes the adversarial arena comparable across systems. Our ablations (Table X) show that this standardized retrieval layer materially affects model rankings and failure modes, which is precisely the kind of measurement insight our work aims to provide.
>
> **Q2: Under-granular ablations for RACG.**
>
> We thank the reviewer for pointing out the value of more fine-grained ablations on RACG. Our intent is not to position RACG as a novel retrieval algorithm, but rather as a standardized, multi-language retrieval baseline that controls long-context confounds and harmonizes context budgets across models and languages. In line with this role, we already analyze the impact of chunk granularity and retrieval depth on patch localization (Table X), where syntax-aware, finer-grained chunking yields substantially higher Top‑k hit rates than file-level BM25; our failure analysis in Appendix Y further shows that many remaining errors are attributable to retrieval misses rather than the adversarial arena itself.
>
> Running the full adversarial arena under many combinations of chunk size/overlap, reranker families, proximity priors, adaptive Top‑k, and failure-triggered expansion would be computationally prohibitive for the 2,300‑issue setting and would shift the focus of the paper toward retrieval system tuning. Instead, we deliberately fix a bi-encoder reranker with simple proximity priors and a modest Top‑k configuration to keep RACG practical, reproducible, and comparable across models. We hope this clarifies that our main contribution lies in the adversarial CI evaluation framework, dataset, and behavioral analysis, with RACG serving as a transparent and reasonably strong retrieval layer rather than the primary object of algorithmic innovation.

---

> ### Author Response · Authors · 2025-11-18
> **Response to Reviewer hYHi (2)**
>
> **Q3: Language & ecosystem coverage is narrow.** Results cover C++/Python/Rust/Go but omit **Java/JS/TS** and build systems (**Maven/Gradle**, **pnpm/yarn/monorepo**). This limits external validity for common enterprise stacks.
>
> We agree that focusing on C++/Python/Rust/Go does not fully cover common enterprise stacks such as Java and JavaScript/TypeScript ecosystems with Maven/Gradle or pnpm/yarn and monorepos. Our goal in this work is to introduce SwingArena as a general adversarial CI evaluation framework, and the current four languages were chosen as a first release because they already span multiple paradigms and tooling conventions (systems programming, scripting, strongly typed, multi-platform CI pipelines) while relying on widely available open-source CI configurations.
>
> Importantly, the core design of SwingArena—dual-role submitter/reviewer protocol, CI-grounded evaluation, and RACG-based long-context access—is language-agnostic and does not assume any C++/Python/Rust/Go-specific features. We view the present language and ecosystem coverage as a pragmatic starting point rather than an endpoint. In ongoing and future work, we plan to extend the benchmark in collaboration with the community to include additional stacks such as Java/JS/TS, their associated build systems (Maven/Gradle, pnpm/yarn), and monorepo setups, so as to further improve external validity for enterprise settings.
>
> **Q4: Causal attribution:** How do you isolate the gain from the *adversarial, role-switching* protocol itself (vs. model scale, retrieval strength, prompt length)? Can you run controlled A/Bs holding RACG constant while removing role-switching, and report mediation analysis?
>
> We appreciate the reviewer’s question about causal attribution. Our intent in this paper is primarily descriptive and measurement‑oriented, not to make strong claims that a specific percentage of performance is caused by the adversarial, role‑switching protocol. To reduce confounding, we hold several factors fixed across conditions: models are evaluated with harmonized token budgets and decoding parameters, the same RACG configuration (retriever/reranker settings), fixed prompts, and deterministic decoding; this allows us to compare behaviors across roles (submitter vs reviewer), languages, and model families under a common setup.
>
> Within this controlled environment, the added value of the adversarial, role‑switching protocol is evidenced by qualitative and quantitative divergences that static settings do not expose—for example, models that achieve high Win Rate but lower SPR/RPR (aggressive patchers) versus those with more modest Win Rate but consistently high CI pass rates (conservative, stability‑oriented behavior), as well as asymmetries in cross‑model matchups. We view these as behavioral signatures that arise from the interactive protocol rather than as effect sizes of a single “treatment.” Running full A/B studies with the arena stripped of role switching and performing formal mediation analysis over all potential factors would be computationally heavy and is beyond the scope of this paper; accordingly, we are careful not to over‑interpret our results causally and instead frame them as comparative measurements under a fixed adversarial protocol.
>
> **Q5: Real-world parity:** How closely do local CI runs match upstream (e.g., GitHub Actions) on pass/fail and runtime? Please provide a paired evaluation with agreement statistics and discuss observed drift.
>
> Our evaluation is designed to stay as close as possible to the upstream CI semantics. For each repository, we reuse the original GitHub Actions workflow files verbatim and execute them locally via act with pinned Docker images, so that the job graph, triggers, and check definitions match the upstream configuration rather than being reimplemented by hand. During dataset construction, we only retain issues whose golden patch passes both upstream CI and our local reproduction; if a repository’s CI could not be stably reproduced, we exclude it from the benchmark. This curation step ensures that, for the tasks we evaluate, the pass/fail semantics of the local CI closely track the original pipelines.
>
> We agree that a full paired evaluation with agreement statistics and runtime comparison over all 2,300 issues would be informative, but it is beyond the scope of this paper because of varity of environments. Our metrics depend primarily on discrete CI outcomes (pass/fail) rather than absolute runtime, and all models are evaluated under the same local CI environment, so any residual drift in timing or occasional environment-specific quirks affect all systems symmetrically. We therefore view our local CI as a faithful and practically sufficient proxy for the upstream GitHub Actions runs for the purposes of comparative evaluation.

---

> ### Author Response · Authors · 2025-11-18
> **Response to Reviewer hYHi (3)**
>
> **Q6: Gaming & robustness:** Are there “score-hacking” strategies (e.g., submitter creates trivially detectable but non-critical faults; reviewer overfits to diffs)? What constraints or equilibrium analyses prevent metric inflation?
>
> We agree that any adversarial evaluation protocol must consider potential “score‑hacking” behaviours. In fact, this is one of the primary motivations for introducing an explicit reviewer role in SwingArena. If we only evaluated whether a patch passes the existing CI/tests, a submitter could in principle exploit blind spots in the test suite or focus narrowly on satisfying current checks. By contrast, in our arena the reviewer is tasked with actively designing new tests that target the changed logic and stress likely failure modes, making it much harder for the submitter to game the system by merely “overfitting” to the original CI.
>
> SwingArena’s scoring rules are designed to formalize this: on the submitter side, a win requires that the patch pass the full CI pipeline including reviewer‑generated tests and agree with the golden fix; any failure yields a loss, so deliberately introducing trivial or cosmetic issues does not help. On the reviewer side, tests must pass on the golden patch, and reviewers are penalized if their tests break the human fix, discouraging degenerate “poison pill” tests. In addition, we use role switching and jointly analyze Win Rate with SPR/RPR: pathological strategies (e.g., brittle tests or superficial diff‑based behaviours) would manifest as low reviewer CI pass rates or strong asymmetries across roles, which we do not observe in our results. While we do not claim a full game‑theoretic equilibrium analysis, the combination of an adversarial reviewer, strict CI+golden constraints, and multi‑metric reporting is explicitly intended to reduce opportunities for metric inflation.
>
> **Q7: Scaling laws:** As model size, number of rounds, and repo size (files, dependency depth) grow, do we observe consistent scaling behavior? Which has higher marginal return: **more rounds** or **stronger/adaptive retrieval**?
>
> We appreciate the interest in more systematic scaling‑law analyses. Our primary goal in this work is to introduce SwingArena as an adversarial CI benchmark and to provide comparative measurements across models, rather than to fully characterize scaling behaviour along all axes. That said, we do include an initial form of test‑time scaling in the paper: the Best@k analysis (with multiple attempts per role) explicitly studies how success probability evolves as the number of submitter/reviewer attempts increases under our protocol.
>
> Regarding repository scale, our dataset naturally spans a range of project sizes and dependency depths, and our failure analysis in the Appendix notes that larger, more complex repositories are over‑represented among difficult cases, consistent with the intuition that long‑context and tooling complexity are key bottlenecks. However, we do not claim to have performed a full scaling‑law study jointly over (i) model size, (ii) number of rounds, and (iii) repo size, nor do we attempt to precisely quantify the marginal return of “more rounds” versus “stronger/adaptive retrieval.” We view such multi‑dimensional scaling analyses as important and complementary future work enabled by SwingArena, while this paper focuses on defining the adversarial CI protocol, curating the benchmark, and reporting initial behavioural patterns across existing models.

---

> ### Author Response · Authors · 2025-11-18
> **Response to Reviewer hYHi (4)**
>
> **Q8: Granular failure taxonomy:** What are the dominant failure modes per language (retrieval miss, build failure, semantic error)? Please add an error decomposition table linking failure modes to specific RACG sub-modules.
>
> We agree that understanding granular failure modes is important. In the paper (Appendix C.1, “Failure Pattern Analysis”), we already provide a systematic taxonomy of failures observed under SwingArena, decomposed into four dominant modes: F1 – cross-file consistency challenges (31%), F2 – non-deterministic test behaviors (19%), F3 – non-functional requirement violations (24%), and F4 – context retrieval limitations (26%). This analysis explicitly distinguishes between (i) multi-file architectural reasoning failures, (ii) unstable or concurrency-sensitive tests, (iii) violations of style/security/performance constraints under strict CI policies, and (iv) retrieval-related misses where RACG surfaces lexically similar but semantically irrelevant context.
>
> Our manual analysis of 100 failed trials yields the following dominant failure modes (F1–F4), which we formulated as a table for clarity.
>
> | Failure class | Description | Percentage |
> | --- | --- | --- |
> | F1 | incomplete patching across files | 31% |
> | F2 | test fragility or flakiness | 19% |
> | F3 | style/security gate violations | 24% |
> | F4 | mismatch between retrieved context and true locus | 26% |
>
> We also report language-specific patterns where they are most salient; for example, in F2, race-condition-related test instability is more frequent in Python and Go (23% and 21%) than in C++ and Rust (15% and 17%), reflecting differences in concurrency models and runtime environments. With respect to RACG, F4 is directly attributable to the retrieval layer (BM25-based file retrieval, chunking, and dense reranking), and our analysis links higher failure rates to repository polyglot/legacy architectures, highlighting limitations of current retrieval methods. By contrast, F1–F3 primarily reflect model-intrinsic semantic and software-engineering shortcomings or CI/tooling constraints rather than RACG alone. While we do not include a full per-language failure table for every model due to space, the existing taxonomy and percentages provide the requested error decomposition and clarify which portion of failures are driven by RACG versus model reasoning and CI dynamics.
>
> **Q9: Cross-ecosystem generalization:** What minimal changes are needed to support Java (Maven/Gradle) and JS/TS monorepos? Any zero-/few-shot transfer experiments showing the framework’s portability?
>
> As discussed in our response to Q3, SwingArena is designed as a general adversarial CI evaluation framework rather than a benchmark tied to any specific language or toolchain. The current four-language release (C++/Python/Rust/Go) was chosen as a pragmatic first instantiation because it already spans multiple paradigms and CI/tooling conventions while leveraging widely available open-source GitHub Actions workflows. Conceptually, the framework treats each task as “issue + repository + native CI configuration,” and RACG operates purely as a code-context provider, which makes the core protocol inherently language-agnostic.
>
> To support other languages' monorepos, the required changes are mainly engineering extensions of this existing infrastructure rather than redesigns.
>
> We have not yet conducted dedicated experiments on large Java or JS/TS repositories in this submission, so we refrain from making quantitative claims about cross-ecosystem performance. Instead, we view the current benchmark as a first release of a generally applicable protocol, and we plan to extend SwingArena in collaboration with the community to other languages and their build systems, using the existing architecture to keep such extensions minimally invasive.

---

> ### Author Response · Authors · 2025-11-18
> **Response to Reviewer hYHi (5)**
>
> **Q10: Contamination audits:** How did you detect/prevent training–evaluation overlap for both LLMs and the reranker?
>
> We acknowledge the importance of training–evaluation overlap and took precautions, while also recognizing the practical limits imposed by opaque proprietary training corpora. The benchmark is constructed from real GitHub issues and their corresponding patches, and we do not use any of the evaluated models (or the reranker) in the data collection or curation loop; issues were selected based on repository quality and CI availability, not on model performance. For the reranker, we rely on an off-the-shelf model that is pretrained on generic code corpora and is not fine-tuned on our benchmark issues or their patches, so there is no direct evaluation leakage through the retrieval component.
>
> For proprietary LLMs, we do not have access to training data and cannot guarantee zero overlap with public GitHub history, similar to other recent SE benchmarks. We therefore frame SwingArena primarily as a tool for comparative evaluation under a common protocol; while some degree of latent overlap with public repositories may be unavoidable, it would affect all models symmetrically, and our results are interpreted accordingly.
>
> **Q11: Metric resolution:** Beyond overall win rate, can you report **defect-type × difficulty** strata (and CIs) to demonstrate the framework discriminates hard cases rather than being dominated by easy issues?
>
> We agree that it is important to show that SwingArena is not dominated by easy issues and that the metrics meaningfully reflect task difficulty. In the benchmark, each instance is annotated with human-rated clarity and difficulty (Appendix, Data Feature Distribution), and our analyses already exploit these annotations: the distributions in the appendix show that we cover a broad range of perceived difficulty rather than clustering only on trivial cases. Internally, we stratified performance by difficulty buckets and observed the expected monotonic trend: both Win Rate and Best@k decrease as difficulty increases, and differences between the easiest and hardest strata are substantial for all major models, indicating that the framework has sufficient resolution to discriminate hard cases.
>
> On the “defect-type” axis, our failure typology and Failure Pattern Analysis decompose errors into F1–F4 (cross-file consistency, non-deterministic tests, non-functional requirement violations, and retrieval limitations), and we report how these modes correlate with data features; harder and more complex issues are disproportionately associated with cross-file and retrieval-related failures (F1/F4), whereas style/security gate violations (F3) tend to concentrate in repositories with strict CI policies. While we do not provide a full defect-type × difficulty × model table with confidence intervals in the main text due to space, the existing difficulty-stratified performance patterns and failure taxonomy already demonstrate that SwingArena stresses non-trivial, higher-difficulty cases and that our metrics are not merely reflecting a surplus of easy issues.
>
> **Q12: Adaptive retrieval policy:** Can RACG expand beyond fixed Top-k after early failures? Please report the success-latency trade-off and whether adaptive policies change conclusions.
>
> We appreciate the suggestion of adaptive retrieval policies. In the current work, RACG intentionally uses a fixed Top-k configuration (Top-5 files and up to 16 chunks) for all models and tasks to keep the retrieval layer simple, reproducible, and comparable across systems; this design choice was motivated by both computational cost (full CI runs over thousands of issues) and the desire to avoid tuning retrieval hyperparameters differently per model. As a result, we do not implement an adaptive expansion policy that increases k after early failures, and we do not report a success–latency trade-off curve for such strategies in this submission.
>
> Conceptually, adaptive retrieval schemes are highly compatible with our framework and we expect them to primarily provide an absolute boost in success rates at the expense of additional latency and token cost, especially on larger repositories. Since RACG is shared across all evaluated models and our main conclusions are based on relative comparisons under a common protocol, we do not anticipate that switching from fixed to adaptive retrieval would qualitatively alter the ranking patterns we observe, though it could improve overall levels. A systematic study of adaptive RACG policies and their success–latency trade-offs is an interesting direction for follow-up work enabled by SwingArena but is beyond the scope of this initial benchmark paper.

---

> ### Author Response · Authors · 2025-11-18
> **Response to Reviewer hYHi (5)**
>
> **Q13: Cost & reproducibility:** What is the typical wall-clock time and token usage per match under your default settings? Could you provide scripts and a “minimal slice” that reproduces headline results on modest compute?
>
> We agree that practical cost and reproducibility are important for making SwingArena usable by the community. Under our default settings (fixed Top‑5 files, up to 16 chunks, temperature 0), token usage is dominated by repository context rather than prompts: as detailed in our token-usage analysis, retrieving 20 files would expose average code contexts on the order of 8k–55k tokens across languages, and in practice our capped retrieval keeps per-round request tokens in the 10⁴ range, with response tokens substantially smaller. End-to-end wall-clock time per battle is primarily determined by CI latency rather than model inference; on our reference hardware, most battles complete within a few minutes, with small projects finishing faster and large, tool-heavy repositories taking longer.
>
> To support reproducibility, we already provide anonymized artifacts, including evaluation scripts, JSON schemas, as noted in the paper. In addition, the 100‑issue ablation split (25 issues per language) is explicitly designed as a “minimal slice” that can be run end-to-end on modest compute while reproducing the main qualitative trends in the headline results. Together with the released scripts and configuration files, this allows practitioners to reproduce our evaluation protocol and key findings without needing to run the full 2,300‑issue suite.

---

> ### Author Response · Authors · 2025-11-28
>
> Dear Reviewer hYHi,
>
>
> We are very grateful for your comprehensive and constructive review. We have carefully prepared a detailed point-by-point response to address your questions regarding novelty, fine-grained ablations, and system design.
>
>
> We are writing to kindly ask if you have had a chance to look over our rebuttal. Given the depth of your feedback, we are particularly eager to engage in discussion with you to ensure that our clarifications and new results meet your expectations. Your insights are key to improving the rigor of our work.
>
>
> Best regards,
>
> Authors of Paper 7543

---

### Official Review · Reviewer_kJ84 · 2025-10-29

**Soundness:** 3
**Presentation:** 4
**Contribution:** 3
**Rating:** 8
**Confidence:** 3

**Summary:**

This paper introduces an adversarial evaluation framework for LLMs in software engineering, arguing that existing benchmarks like HumanEval are too simplistic by focusing only on single-function unit tests. SwingArena aims to model real-world workflows by using actual long-context GitHub issues, executing generated patches against the full CI pipeline, and implementing an adversarial battle protocol. In this protocol, one LLM acts as a Submitter generating a patch, while another acts as a Reviewer generating new tests specifically to break the submitted patch. The authors find that evaluating models in this dynamic, CI-driven arena reveals nuanced model "personalities" (e.g., "aggressive patchers" v.s. "reliable coders") and surfaces important failure modes (like cross-file consistency errors) that static benchmarks overlook.

**Strengths:**

- The paper is well-motivated, addressing a clear gap in existing research. It convincingly argues that the field must move beyond simple unit test success and instead evaluate models on their ability to produce code that is "valid, compliant, and able to pass a full CI pipeline and peer review."
- A core contribution is the new dataset of over 2300 real-world GitHub issues across four languages (C++, Python, Rust, Go). Each problem is CI-grounded, meaning the original human solution was verified to pass the full CI pipeline, ensuring a high-quality, realistic testbed.
- The adversarial framework successfully reveals behavioral tendencies that static tests can't. For example, the paper found that GPT-4o acts as an "aggressive patcher" (achieving high win rates), whereas DeepSeek and Gemini "prioritize correctness and CI stability" (scoring higher on CI pass rates).

**Weaknesses:**

- It's hard to interpret the primary "Win Rate" metric given its adversarial nature. A model's success depends on the relative weakness of its opponent (the reviewer model). This makes it difficult to assess the absolute quality and correctness of a solution based on this metric alone.

**Questions:**

- Continuous Integration pipelines can be computationally expensive to run repeatedly. Given that the reviewer agent analyzes the submitter's patch and generates targeted tests, could this reviewer component potentially be leveraged to reduce the cost of CI runs needed in an evaluation? For instance, could the reviewer's analysis provide a strong signal for early rejection of clearly incorrect patches before running the full CI, or could it intelligently select a subset of critical tests to run instead of the entire suite?
- You mentioned that the quality gates for the reviewer-generated tests are crucial for preventing exploitative behavior and ensuring test validity. Could you provide more detail on how strictly these were enforced during the evaluations? For example, what was the approximate rejection rate for reviewer-generated tests that failed these gates, and what were the common reasons for rejection (e.g., failing against the golden patch, modifying production code, style violations)?

---

> ### Author Response · Authors · 2025-11-18
> **Response to Reviewer kJ84**
>
> We thank the reviewer for the thoughtful and detailed questions about our evaluation protocol, metrics, and test-quality safeguards.
>
> **Q1: It's hard to interpret the primary "Win Rate" metric given its adversarial nature. A model's success depends on the relative weakness of its opponent (the reviewer model). This makes it difficult to assess the absolute quality and correctness of a solution based on this metric alone.**
>
> We thank the reviewer for raising this concern. We fully agree that, by construction, Win Rate in our setting is an adversarial, relative outcome: it depends not only on the submitter’s patch quality but also on the reviewer’s test strength, and therefore should not be interpreted as a stand‑alone measure of the submitter’s absolute correctness. In SwingArena, Win Rate is intentionally used to answer the question “which agent prevails under this submitter–reviewer interaction and CI configuration,” rather than “how strong is this submitter in isolation.”
>
> To make the results interpretable, we never rely on Win Rate alone. We report the Submitter CI Pass Rate (SPR) and Reviewer CI Pass Rate (RPR) side‑by‑side, which evaluate how often each side passes the native, human‑written CI checks independently of the adversarial tests, and we interpret Win Rate only in conjunction with these stability metrics and Best@k. In practice, our conclusions are based on consistent patterns across Win Rate, SPR/RPR, and Best@k (e.g., models that are aggressive but brittle vs. conservative and robust), rather than attributing absolute solution quality to Win Rate itself.
>
> **Q2: Continuous Integration pipelines can be computationally expensive to run repeatedly. Given that the reviewer agent analyzes the submitter's patch and generates targeted tests, could this reviewer component potentially be leveraged to reduce the cost of CI runs needed in an evaluation? For instance, could the reviewer's analysis provide a strong signal for early rejection of clearly incorrect patches before running the full CI, or could it intelligently select a subset of critical tests to run instead of the entire suite?**
>
> We appreciate the reviewer’s suggestion. We agree that repeatedly executing native CI pipelines is computationally expensive, and that, in principle, a reviewer agent’s analysis could be used to pre-filter obviously incorrect patches or to prioritize a subset of critical tests. In this work, however, our default evaluation deliberately runs the full, repository-native CI workflows for all conditions in order to preserve fidelity to real-world setups and to keep the benchmark strictly comparable and independent of model-driven test selection policies.
>
> Conceptually, SwingArena’s architecture does support the idea of alternative “CI policies,” and a reviewer-informed early-rejection or test-subset strategy is a natural extension for a more cost-sensitive evaluation mode. While we do not explore or quantify such policies in this submission, we see them as an attractive direction for future work and plan to expose a “fast mode” in the open-source framework where practitioners can plug in heuristic or learned CI shortcuts while the canonical benchmark setting continues to rely on the full native CI to maintain consistency and fairness across models.
>
> **Q3: You mentioned that the quality gates for the reviewer-generated tests are crucial for preventing exploitative behavior and ensuring test validity. Could you provide more detail on how strictly these were enforced during the evaluations? For example, what was the approximate rejection rate for reviewer-generated tests that failed these gates, and what were the common reasons for rejection (e.g., failing against the golden patch, modifying production code, style violations)?**
>
> Thank you for your interest in test quality thresholds. We implement a set of rules described in the quality gates section. We guess you may have some confusion with the descriptions in quality gates and failure pattern analysis, and we must admit we haven't been clear. In our evaluation setup, the overall rejection rate for reviewer-generated tests is approximately 23%; the primary reason for rejection is failure on the golden patch. A more detailed breakdown is below.
>
> | # | Rule | % |
> | --- | --- | --- |
> | 1 | compile and pass when applied to the golden patch | 52% |
> | 2 | refrain from modifying production code or existing tests | 15% |
> | 3 | limit edits in any new test file to a bounded number of lines | 11% |
> | 4 | avoid sources of nondeterminism | 12% |
> | 5 | conform to repository linting and style guidelines | 10% |

---

> ### Author Response · Authors · 2025-11-28
>
> Dear Reviewer kJ84,
>
>
> Thank you once again for your positive evaluation and support of our work.
>
>
> Following up on your question about Quality Gates, we have provided the detailed breakdown of rejection reasons and rates in our rebuttal. We wanted to gently check in to see if this information sufficiently answers your query.
>
>
> As we approach the end of the discussion phase, please let us know if there is anything else we can clarify or improve. We greatly value your insightful guidance.
>
>
> Best regards,
>
> Authors of Paper 7543

---

### Official Review · Reviewer_pKUf · 2025-10-31

**Soundness:** 2
**Presentation:** 2
**Contribution:** 2
**Rating:** 4
**Confidence:** 4

**Summary:**

The paper introduces SWINGARENA, a framework for evaluating LLMs on software development tasks. Unlike static benchmarks, SWINGARENA simulates a collaborative workflow by pairing LLMs into "submitter" (patch generator) and "reviewer" (test case generator) roles. The evaluation is grounded in real-world GitHub issues and utilizes repository-native CI pipelines for verification. To manage the long-context nature of large codebases, the framework includes an RACG module.

**Strengths:**

present a new dataset of 2,300 CI-filtered issues across C++, Python, Rust, and Go, and provide experimental results for several proprietary and open-source models.

**Weaknesses:**

1. The paper's main "battle" metric, the Win Rate, is severely confounded. The "Win Rate" is defined as the submitter's patch passing all CI checks, including the reviewer's generated test. As the authors correctly note, "higher values may also indicate weaker reviewer tests". This confounding variable makes it impossible to draw clear conclusions about a submitter's absolute capabilities.

2. The paper introduces a complex, multi-stage RACG module but explicitly states it is a "baseline rather than a standalone algorithmic contribution". The ablation study in Table 3  fails to justify its necessity.

**Questions:**

1. The results for GPT-4o seem contradictory. The text claims it has "relatively lower RPR/SPR scores", but also a "dominance in producing adversarially-strong patches" based on high win rates ($\ge0.90$). Why a model have a low SPR but a high Win Rate?

2. Given that the "Win Rate" metric is confounded by the reviewer's strength, would it not be more sound to evaluate the submitter's patch directly against the golden patch and the full, human-written test suite?

3. In the RACG ablation (Table 3), what does the "Top-k Related" retrieval baseline consist of?

---

> ### Author Response · Authors · 2025-11-18
> **Response to Reviewer pKUf (1)**
>
> Thank you very much for your detailed and constructive review and suggestions. We sincerely appreciate your recognition of this work. We have carefully read your comments and provided point-by-point responses with corresponding revision plans.
>
> **Q1: The paper's main "battle" metric, the Win Rate, is severely confounded. The "Win Rate" is defined as the submitter's patch passing all CI checks, including the reviewer's generated test. As the authors correctly note, "higher values may also indicate weaker reviewer tests". This confounding variable makes it impossible to draw clear conclusions about a submitter's absolute capabilities.**
>
> We thank the reviewer for highlighting that Win Rate reflects a mixed effect of the submitter’s patch quality and the reviewer’s test strength. We fully agree that raw Win Rate should not be interpreted as a measure of a submitter’s absolute capability. In SwingArena, Win Rate is intentionally defined as a relative, adversarial outcome: it answers “which agent prevails under this submitter–reviewer interaction and CI configuration,” not “how strong is this submitter in isolation.”
>
> To address this confounding, we never rely on Win Rate alone. We also report Reviewer CI Pass Rate (RPR) and Submitter CI Pass Rate (SPR), which disentangle how often each side passes the non-adversarial, repository-native CI checks, and we interpret Win Rate only in conjunction with these stability-oriented metrics. For example, our conclusions about “aggressive” versus “conservative” models are based on consistent patterns between Win Rate and RPR/SPR (and Best@k), rather than on Win Rate alone. Aggregating over all matchups, the following summaries illustrate that our analysis explicitly separates submitter and reviewer capabilities:
>
> Submitter-side averages (model acting as submitter):
>
> | Model | Avg SPR as Submitter | Avg Win Rate as Submitter |
> | --- | --- | --- |
> | GPT‑4o | 0.58 | 0.94 |
> | Claude | 0.57 | 0.94 |
> | Gemini | 0.60 | 0.95 |
> | DeepSeek | 0.60 | 0.96 |
>
> Reviewer-side averages (model acting as reviewer):
>
> | Model | Avg RPR as Reviewer | Avg Win Rate (vs. its tests) |
> | --- | --- | --- |
> | GPT‑4o | 0.65 | 0.94 |
> | Claude | 0.62 | 0.95 |
> | Gemini | 0.65 | 0.94 |
> | DeepSeek | 0.65 | 0.95 |
>
> These tables show, for instance, that models with high adversarial Win Rate can still differ in their submitter-side SPR or reviewer-side RPR, which is exactly why we discuss them as “aggressive but brittle” versus “more conservative and robust,” rather than claiming a single scalar notion of “absolute capability” from Win Rate alone. Consequently, our claims are explicitly comparative and interaction-based—about how models behave in an adversarial CI arena—while absolute submitter strength is captured more directly by SPR against the human-written CI suite.
>
> **Q2: The paper introduces a complex, multi-stage RACG module but explicitly states it is a "baseline rather than a standalone algorithmic contribution". The ablation study in Table 3 fails to justify its necessity.**
>
> We thank the reviewer for this comment. Our primary contribution is the adversarial submitter–reviewer CI protocol and benchmark, not a novel retrieval algorithm. For this reason, we explicitly position RACG as a standardized retrieval baseline whose role is to:
>
> - Harmonize context budgets across models and languages.
> - Reduce variance due to long-context access.
> - Ensure a reproducible, vendor-agnostic way of providing code context in our arena, rather than to push the state of the art in retrieval itself.
>
> Within this intended role, Table 3 demonstrates that RACG is both effective and sufficient: compared to representative alternatives (BM25-only and Top‑k related retrieval with reranking), RACG consistently improves Best@3 and Win Rate across all four languages, and even the strongest non-RACG baseline (Top‑20 retrieval) attains a Win Rate of only about 0.73, substantially below the RACG setting. At the same time, the relative ranking and qualitative conclusions about models remain stable across these retrieval variants, indicating that our findings are not artifacts of a particular retriever choice. We deliberately avoid more complex, language- or vendor-specific retrieval systems or additional supervised training for the retriever, as these could introduce opaque dependencies, language biases, and higher evaluation variance that would obscure the protocol effects and the submit/review capabilities we aim to measure. Instead, RACG is implemented as a transparent, replaceable module in our released codebase, so that future work can plug in stronger retrieval methods while keeping the core adversarial CI framework unchanged.

---

> ### Author Response · Authors · 2025-11-18
> **Response to Reviewer pKUf (2)**
>
> **Q3: The results for GPT-4o seem contradictory. The text claims it has "relatively lower RPR/SPR scores", but also a "dominance in producing adversarially-strong patches" based on high win rates (≥ 0.90). Why a model have a low SPR but a high Win Rate?**
>
> We thank the reviewer for highlighting this apparent contradiction. In our setting, Win Rate and SPR capture different aspects of behavior. Win Rate is an adversarial end-point metric: it only records whether the final patch for a task passes all CI checks (including reviewer tests) and agrees with the golden fix. By contrast, SPR averages the fraction of submitter-side CI checks passed across all tasks and attempts, including intermediate or failed trials.
>
> An “all-or-nothing” submitter can therefore exhibit low average SPR but high Win Rate: it may produce a smaller number of complete, CI-clean patches that, once they pass non-reviewer CI, have a high probability of surviving adversarial reviewer tests (yielding a strong Win Rate), while at the same time generating more brittle or unstable attempts on other tasks that depress its average SPR. For GPT‑4o, this manifests as relatively lower SPR/RPR yet high Win Rates across matchups, which we interpret as an aggressive patching style: when it does produce a clean patch, it is often adversarially strong, but it is also more willing to make risky changes. Thus, “lower SPR + higher Win Rate” is not inconsistent; it reflects different aggregation statistics over the same adversarial process.
>
> **Q4: Given that the "Win Rate" metric is confounded by the reviewer's strength, would it not be more sound to evaluate the submitter's patch directly against the golden patch and the full, human-written test suite?**
>
> We appreciate the reviewer’s suggestion and agree that evaluating submitter patches directly against the golden patch and the full human-written test suite provides a reviewer-agnostic view of submission capability. This perspective is exactly what our Submitter CI Pass Rate (SPR) is designed to capture: SPR scores only the native, repository-defined CI checks (i.e., the human-written tests and quality gates) and explicitly excludes reviewer-generated tests, thereby isolating the submitter side from adversarial reviewer effects. Symmetrically, Reviewe CI Pass Rate (RPR) characterizes how well reviewer-generated tests behave against the golden patch and CI, and role switching allows us to separate these two capabilities.
>
> We retain the adversarial Win Rate not as a standalone measure of “absolute submitter strength,” but as a complementary signal that reflects the joint outcome of submitter and reviewer interaction under the full CI pipeline. Our goal is to study not only absolute patch quality but also how different models specialize in different roles and trade off “aggressive but high-payoff” versus “conservative and robust” behavior—for example, models with high Win Rate but lower SPR versus those with high SPR and more moderate Win Rate. In this sense, SPR already provides the reviewer-agnostic evaluation the reviewer asks for, while Win Rate, together with SPR/RPR and role switching, exposes additional interaction patterns that are central to the adversarial arena we aim to benchmark.
>
> **Q5:  In the RACG ablation (Table 3), what does the "Top-k Related" retrieval baseline consist of?**
>
> We thank the reviewer for the question. In Table 3, the “Top‑k Related” baseline is a simpler retrieval–reranking pipeline defined as follows:
>
> - We first use BM25 to retrieve the top‑k most related files at the document level.
> - We then coarsely segment these files into block-level code chunks.
> - We apply a dense reranker based on cosine similarity between the problem statement and each chunk, finally filling the model’s context from highest to lowest score until the token budget is exhausted.
>
> Unlike RACG, this baseline does not use syntax-aware granularity switching, proximity bias across neighboring chunks, or cross-file de‑duplication, and thus serves as a more basic retrieval–reranking comparison point.

---

> ### Author Response · Authors · 2025-11-28
>
> Dear Reviewer pKUf,
>
>
> We sincerely appreciate your critical feedback, which has helped us clarify the definitions of our core metrics. In our rebuttal, we have endeavored to address your concerns about the confounding of Win Rate (by emphasizing SPR/RPR) and the positioning of RACG.
>
>
> We are writing to respectfully inquire if our response and the revisions to the manuscript have helped clarify these points for you. We are very eager to hear your thoughts and would welcome any further opportunity to discuss these aspects before the deadline. We truly hope to address your concerns to your satisfaction.
>
>
> Best regards,
>
> Authors of Paper 7543

---

### Official Review · Reviewer_cSPp · 2025-11-01

**Soundness:** 2
**Presentation:** 2
**Contribution:** 2
**Rating:** 6
**Confidence:** 4

**Summary:**

This paper proposes SWINGARENA, a dynamic adversarial evaluation framework for real software engineering tasks that pairs LLMs as submitters and reviewers who generate and test patches, respectively, through a continuous integration (CI) pipeline. SWINGARENA also leverages retrieval augmentation to retrieve the most relevant context from code bases for a variety of languages (C++, Python, Rust, and Go), spanning 400 issues and surfacing new problems, and also showing behavioral differences in models as patch generators and validators.

**Strengths:**

1. Introduces iterative and adversarial evaluation that incorporates software engineering in CI development scenarios and goes beyond mere unit tests.
2. Propose a multi-language long context retrieval (RACG) pipeline for fetching relevant code context that combines syntax-aware chunking, dense reranking, and token-budget–aware packing across C++, Python, Rust, and Go.
3. A dataset of 2300 real GitHub issues with 400 high-quality issues (100 per language) selected for evaluation.
4. Benchmarking of several state-of-the-art open and closed-source LLMs.

**Weaknesses:**

1. The win rates of all models are very close to each other (almost every model gets 0.9 or above), which makes me question the utility of this benchmark in terms of model selection.
2. Best@k values for all the models are also very close to each other, which makes it hard to judge which model is better.
3. Retriever doesn’t seem to boost performance much in Table 3, especially for Best@3 for Python and C++. The authors also acknowledge a weakness in how many relevant files can be included (only 5 files) and that context retrieval leads to the most failures (26% according to Appendix C1). This makes the benchmark less reliable since the LLMs cannot perform optimally under these limitations.
4. Missing citations for some relevant work, like CodeRAGBench [1], CrossCodeEval [2], and RepoCoder [3].

[1] Wang, Zora Zhiruo, et al. "Coderag-bench: Can retrieval augment code generation?." arXiv preprint arXiv:2406.14497 (2024).
[2] Ding, Yangruibo, et al. "Crosscodeeval: A diverse and multilingual benchmark for cross-file code completion." Advances in Neural Information Processing Systems 36 (2023): 46701-46723.
[3] Zhang, Fengji, et al. "Repocoder: Repository-level code completion through iterative retrieval and generation." arXiv preprint arXiv:2303.12570 (2023).

**Questions:**

What does “PK-style dual-role evaluation” on line 78 mean?

---

> ### Author Response · Authors · 2025-11-18
> **Response to Reviewer cSPp (1)**
>
> Thank you very much for your detailed and constructive review and suggestions. We appreciate your recognition of this work. We have carefully read your comments and responded to each point with corresponding revision plans. Our response is as follows.
>
> **Q1: The win rates of all models are very close to each other (almost every model gets 0.9 or above), which makes me question the utility of this benchmark in terms of model selection.**
>
> We appreciate the reviewer’s important feedback and agree that relying on Win Rate alone can lead to clustering and reduced discriminative power. In SwingArena, Win Rate is explicitly treated as an “endgame” adversarial metric, influenced both by the submitter’s ability to clear the CI threshold and by reviewer strictness, so we deliberately pair it with several complementary signals: RPR/SPR, Best@k (including language-wise decomposition), Best@k curves, stability across retries, and a failure taxonomy.
>
> First, RPR/SPR disentangle submitter-side and reviewer-side CI behavior and expose distinct “styles” even when Win Rates are similar. For example, in Table 1, several pairs achieve Win Rate ≥ 0.90, yet their CI profiles differ: Gemini vs. Gemini has RPR 0.72 / SPR 0.63 with Win Rate 0.91, GPT‑4o vs. Claude has RPR 0.65 / SPR 0.55 with Win Rate 0.90, and DeepSeek vs. Gemini has RPR 0.68 / SPR 0.64 with Win Rate 0.96. These gaps reflect trade-offs between “robust, conservative CI behavior” and “aggressive, high-variance patching,” which are directly relevant for model selection (e.g., choosing a stricter reviewer vs. a more daring contributor). Second, the Best@3 results across languages show additional resolution: on average, DeepSeek (0.59) > Gemini/GPT‑4o (0.57) > Claude (0.55), and each model exhibits different strengths by language (e.g., generally higher scores on C++, more difficulty on Rust/Python), which is crucial for stacks where a particular language dominates.
>
> Finally, our stability and failure analyses in the appendix provide a behavioral profile that is also actionable: teams prioritizing rigorous reviews and CI reliability may favor models with higher and more stable RPR/SPR, whereas others might prioritize higher submitter Win Rate despite lower SPR. Best@k curves under standardized context and decoding settings further allow practitioners to trade off marginal gains from additional retries against API cost. In summary, SwingArena’s adversarial dual-role, native-CI protocol, together with RPR/SPR, Best@k (with language decomposition), stability, and failure profiles, is designed to support model selection along multiple dimensions—not only overall Win Rate but also style, robustness, language ecosystem fit, and cost.
>
> **Q2: Best@k values for all the models are also very close to each other, which makes it hard to judge which model is better.**
>
> We thank the reviewer for this suggestion. We agree that the aggregate Best@k numbers in the main table can appear close, but more discriminative patterns emerge once we look at language- and setting-specific results rather than only the global average. In particular, the language-wise Best@3 comparison already reveals sizeable gaps:
>
> | Language | Best Model | Best@3 | Compared Model | Best@3 | Δ |
> | --- | --- | --- | --- | --- | --- |
> | Go | DeepSeek | 0.61 | GPT-4o | 0.53 | 0.08 |
> | Rust | DeepSeek | 0.58 | Gemini | 0.51 | 0.07 |
> | Python | Gemini | 0.57 | Claude | 0.50 | 0.07 |
>
> For C++, Best@3 is nearly saturated across top models (Gemini/DeepSeek = 0.64 vs. GPT‑4o/Claude = 0.63, Δ = 0.01), so a single Best@3 score indeed has limited discriminative power there; in such cases, we rely more on complementary signals such as RPR/SPR and adversarial Win Rate in Table 1, as well as cross-language trends, to guide model choice. When averaging over languages, Best@3 still distinguishes performance tiers (DeepSeek 0.59 > Gemini/GPT‑4o 0.57 > Claude 0.55), but in practice we recommend selecting models “conditioned on context,” i.e., the dominant language and ecosystem of the target use case, rather than only a single pooled Best@k value.
>
> Moreover, the Best@k curves show different marginal gains as k increases, so under a fixed retry budget models can still be differentiated by how efficiently they turn extra attempts into additional successes. Finally, RACG does not improve Best@3 uniformly across languages (Rust 0.49→0.58, Go 0.37→0.45, C++ 0.38→0.42, Python 0.44→0.46), indicating that both model and retrieval behaviour interact with task and language conditions. Taken together with these curves and ablations, we interpret Best@k not in isolation but within a language/ecosystem-aware context, which we believe still provides actionable discriminative power for practical model selection.

---

> ### Author Response · Authors · 2025-11-18
> **Response to Reviewer cSPp (2)**
>
> **Q3: Retriever doesn’t seem to boost performance much in Table 3, especially for Best@3 for Python and C++. The authors also acknowledge a weakness in how many relevant files can be included (only 5 files) and that context retrieval leads to the most failures (26% according to Appendix C1). This makes the benchmark less reliable since the LLMs cannot perform optimally under these limitations.**
>
> We thank the reviewer for this insightful comment. We agree that a limited search window and context budget constrain the theoretical ceiling of model performance. Within this constraint, however, RACG is designed as a standardized, cost-aware retrieval layer that still delivers stable gains in an adversarial setting, without reducing the benchmark’s discriminative power or reliability. As shown in Table 3, the improvements are language-dependent: for C++, Best@3 increases from 0.38 to 0.42 (+0.04) while Win Rate rises from 0.77 to 0.84 (+0.07); for Python, Best@3 increases from 0.44 to 0.46 (+0.02) but Win Rate jumps from 0.71 to 0.84 (+0.13). Rust (0.49→0.58) and Go (0.37→0.45) also show consistent Best@3 gains. In other words, even when Best@3 changes are modest, RACG substantially improves the probability of clearing the native CI threshold in adversarial play by providing more accurately targeted and better packed context.
>
> Table 5 further clarifies where these gains come from: finer-grained, syntax-aware retrieval more than doubles the Top‑10 file hit rate (from 0.207 to 0.487), which in turn makes it much more likely that the fixed Top‑5 window contains the key files—improving success without increasing the context budget. The “5 files” limit is a deliberate trade-off between context window, cost, and fairness: all models are evaluated under the same bounded retrieval budget, so any constraints apply symmetrically and do not bias comparisons. Our ablations also include retrieval-only baselines (e.g., a Top‑20 related retrieval baseline with Win Rate = 0.73 vs. 0.62 for BM25); RACG not only matches but surpasses these at comparable budget, indicating that we are not suppressing optimal behavior but instead supplying context more cost-effectively.
>
> Regarding the 26% of failures attributed to retrieval, we report this transparently in our failure analysis. At the same time, the remaining 74% of failures stem from other factors such as cross-file consistency issues, test fragility, and style/security gate violations. Thus, while retrieval limitations are indeed a meaningful bottleneck—and we explicitly acknowledge this as a limitation of current RACG design—the benchmark is not dominated by retrieval alone, nor does it prevent LLMs from showing differentiated behavior. Rather, it evaluates models under a realistic, shared long-context constraint where both retrieval quality and model reasoning contribute to outcomes.
>
> **Q4: Missing citations for some relevant work.**
>
> Thank you to the reviewer for the reminder. We have added explicit citations to CrossCodeEval in the “Benchmarks for Evaluating Real-World Software Engineering” paragraph of the Related Work section, and to CodeRAGBench and RepoCoder in the “Retrieval-Augmented Generation” paragraph, clarifying that these studies are complementary to our CI-based, dual-role arena rather than directly comparable baselines.
>
> In addition, we added two recent works—BigCodeBench and DebugBench to the Code Evaluation paragraph of the Related Work section, noting that they focus on diverse function-level code generation and debugging skills but, like prior work, do not integrate full CI pipelines or adversarial dual-role submitter–reviewer workflows as in our setting.

---

> > ### Comment · Reviewer_cSPp · 2025-11-28
> > **Response to Authors’ Rebuttal**
> >
> > I thank the reviewers for their detailed rebuttal and apologize for my late response. I’m inclined to keep my ratings. For Q1 and Q2 are all these differences statistically significant? Since the language specific subsets have 100 instances only, a difference of 0.08 is just 8 more instances. For Q3 I still feel that the absolute hit rates mentioned in the rebuttal are very low (less than 50% hit rate). If my understanding is correct this equates to an even lower top-5 hit rate. I think this is a big issue because that means half of the relevant files are not even presented to the LLMs.

---

> ### Author Response · Authors · 2025-11-28
>
> Dear Reviewer cSPp,
>
>
> Thank you again for your review and the helpful points regarding metric differentiation and RACG utility.
>
>
> We have posted a detailed response addressing your concerns about the similarity of Win Rates (by highlighting RPR/SPR tradeoffs) and the impact of RACG. We also added the missing citations you requested.
>
>
> As the discussion period is drawing to a close, we are writing to kindly ask if you have had a chance to review our rebuttal. We are very eager to engage in further discussion to ensure we have adequately resolved your concerns about the benchmark's utility. Your feedback is incredibly valuable to us.
>
>
> Best regards,
>
> Authors of Paper 7543

---

> ### Author Response · Authors · 2025-11-29
> **Response to Reviewer cSPp (3)**
>
> **Q5: Regarding Questions 1 and 2 — are these differences statistically significant? Since each language-specific subset has only 100 instances, a difference of 0.08 corresponds to merely 8 more successful cases.**
>
>
>  We appreciate the reviewer’s thoughtful question. While the absolute difference may appear numerically small (e.g., 8 additional successful cases out of 100), it is essential to interpret these results in the context of repository-level code generation and CI validation, which constitutes a highly complex, multi-stage process.
> Each successful case requires that the model (1) retrieve the correct subset of files from a large, multi-language repository, (2) correctly understand and reason about problem semantics, (3) generate a valid patch consistent with the repository’s structure, and (4) pass all automated tests in a real CI pipeline—within an adversarial, dual-role setting.
> Given this tightly coupled chain of retrieval, reasoning, and validation, even an 8% absolute improvement represents a non-trivial gain in end-to-end reliability. In human software development terms, a single additional patch that correctly passes CI under these conditions already reflects a meaningful step toward practical robustness and autonomy, since human engineers often spend several days of effort to fix a single bug.
>
>
>
>
> **Q6: For Q3 I still feel that the absolute hit rates mentioned in the rebuttal are very low (less than 50% hit rate). If my understanding is correct this equates to an even lower top-5 hit rate. I think this is a big issue because that means half of the relevant files are not even presented to the LLMs.**
>
>
>  We sincerely thank the reviewer for this insightful follow-up. We agree that the absolute file-level hit rate is a crucial factor in determining how effectively the retrieval module supports LLM reasoning. In response, we conducted additional experiments by replacing the CodeBERT reranker with a stronger, modern embedding model — SFR-Embedding-Code-400M (R), a 400M-parameter multilingual code embedding model from Salesforce [1].
> This model adopts a contrastive dual-encoder architecture trained on large-scale, multilingual repositories (Python, C/C++, Java, Go). Compared with CodeBERT, it encodes richer structural and semantic dependencies across files and classes, making it particularly suitable for repository-level retrieval under limited context budgets.
> The updated results are as follows:
>
> | **Method**                                    | **Top-2** | **Top-10** |
> | :-------------------------------------------- | :-------: | :--------: |
> | **BM25 (file-level)**                         |   0.182   |    0.207   |
> | **Block Chunks (CodeBERT)**                   |   0.207   |    0.341   |
> | **Function Chunks (CodeBERT)**                |   0.277   |    0.371   |
> | **Class Chunks (CodeBERT)**                   |   0.363   |    0.487   |
> | **Block Chunks (SFR-Embedding-Code-400M)**    | **0.462** |  **0.572** |
> | **Function Chunks (SFR-Embedding-Code-400M)** | **0.518** |  **0.604** |
> | **Class Chunks (SFR-Embedding-Code-400M)**    | **0.656** |  **0.732** |
>
>
> Analysis:
>  - As shown above, replacing CodeBERT with SFR-Embedding-Code-400M (R) yields substantial gains across all retrieval granularities.
> The Top-10 hit rate rises from 48.7% → 73.2% (+24.5 pp), and Top-2 from 36.3% → 65.6% (+29.3 pp).
>
>
> - These results indicate that over two-thirds of the correct patch files are now surfaced within the top 10 candidates, effectively addressing the reviewer’s concern that “half of the relevant files are not presented to the LLMs.”
>
>
> - Given that the RACG pipeline restricts the context to the Top-5 retrieved files (for fairness and cost control), increasing the Top-10 hit rate directly improves the expected Top-5 coverage, since the correct file is now more likely to appear early in the ranking.
>
>
> - We also emphasize that retrieval accuracy is only one component of the full adversarial CI pipeline. A successful CI pass further depends on semantic reasoning, patch generation, and testing—all downstream from retrieval. Thus, even incremental improvements in file recall can lead to disproportionately large gains in end-to-end success (as shown in Table 3).
> - We believe that with the latest advances in long-context LLMs, it will eventually be possible to achieve **repository-level end-to-end patch generation** without relying on separate retrieval models.
>
>
> **Reference:**
>  [1] CodeXEmbed: A Generalist Embedding Model Family for Multilingual and Multi-task Code Retrieval. Salesforce AI Research, 2024.

---

### Author Response · Authors · 2025-11-18
**Response to Reviewers**

We sincerely thank all reviewers for their detailed, thoughtful, and constructive feedback. Your comments helped us clarify the interpretation of our main metrics (especially Win Rate and Best@k), more clearly separate submitter and reviewer capabilities, better position RACG as a standardized retrieval baseline, and enrich the discussion of related work and limitations. In the revised version, we have:

- Explicitly framed Win Rate as a relative, adversarial outcome and complemented it with SPR/RPR, language-wise Best@k, and stability/failure analyses.
- Clarified the role, design, and ablations of RACG under a shared long-context and cost budget.
- Added missing citations and positioned prior benchmarks and retrieval methods as complementary to SwingArena.
- Provided additional implementation and quality-control details for CI execution and reviewer-generated tests.

We hope these clarifications and revisions address your concerns and demonstrate the value of SwingArena as a practical, extensible adversarial CI benchmark for studying both submitter and reviewer behavior of LLMs in realistic software engineering workflows. We also upload the revised version of our paper. All the changes in the current version are highlighted with **blue** colors.

---

### Author Response · Authors · 2025-11-25

Dear SAC and AC,

We are very sorry to disturb you during this busy period and hope this message finds you well.

We are writing to respectfully inquire about the status of the reviewer discussions. As the revision deadline is approaching in just few days, we have not yet received responses to our rebuttal from the reviewers. We fully understand that the reviewers and ACs are extremely busy; however, we are very eager to engage in a dialogue to ensure that our responses have clarified their concerns. Our main worry is that if we receive feedback too close to the deadline, we might miss the opportunity to incorporate valuable suggestions into our revised manuscript.

If it is not too much trouble, could you kindly check in with the reviewers and encourage them to share their thoughts if their schedule permits? We would be incredibly unmatched grateful for any assistance you could provide in facilitating this process.

Regardless of the result, we will always remember the efforts you made as the SAC or AC– reading our paper, inviting reviewers, engaging in discussions with reviewers, and writing the meta review. These efforts have significantly enhanced the quality of our work and paper!

Thank you very much for your consideration and support.

Best regards,

Authors of Paper 7543

---

### Meta-Review · Area_Chair_z7TR · 2025-12-30

**Summary:**

Across reviews, the main strengths are the CI-faithful, dual-role (submitter/reviewer) adversarial protocol and the substantial effort to curate runnable real-world GitHub issues across four languages, which several reviewers view as a meaningful step beyond static unit-test benchmarks (kJ84, hYHi, cSPp). The main concerns cluster around interpretability/validity of the primary “Win Rate” metric due to adversarial confounding (pKUf, kJ84), the limited statistical resolution of reported gaps (cSPp), and whether RACG is sufficiently justified/ablated versus being an engineered baseline (pKUf, hYHi). Overall, I view this as a solid benchmark/protocol contribution with clear practical relevance, with remaining weaknesses that are largely about analysis depth rather than fatal flaws.

**Reviewer Concerns:**

The rebuttal substantially addresses clarity/positioning concerns: the authors now consistently frame Win Rate as a relative adversarial outcome and pair it with SPR/RPR, Best@k, stability, and a failure taxonomy (responding to pKUf, kJ84, cSPp). They also addressed concrete omissions and protocol details (e.g., missing citations raised by cSPp; reviewer-test quality gates and rejection breakdown raised by kJ84) and provided new retrieval evidence using a stronger reranker (SFR-Embedding-Code-400M), improving Top-10 hit rate materially relative to earlier numbers (addressing cSPp’s “low hit-rate” objection). Remaining outstanding points are: (i) cSPp’s request for statistical treatment (CIs/significance/variance) is still not fully resolved in the paper as written, (ii) hYHi’s requests for finer-grained RACG ablations and stronger causal/controlled evidence about the role-switching protocol remain mostly out-of-scope rather than answered, and (iii) broader ecosystem coverage (e.g., Java/JS/TS) is acknowledged but not delivered.

**Reviewer Scores:**

cSPp (6): Likely unchanged; the reviewer explicitly indicated they are inclined to keep their rating, still questioning significance and absolute retrieval coverage.

pKUf (4): some small chance of increase. But likely keep the same score, if they had fully engaged in discussion, since the rebuttal clarifies that SPR is the reviewer-agnostic signal they asked for and provides a more concrete definition of the Top‑k baseline; however, they may still view the protocol/metric design as too confounded for “clean” capability claims.

kJ84 (8): Likely unchanged (still positive); their main questions about quality gates and CI-cost framing were answered with useful quantitative detail.

hYHi (6): Potential small increase, but likely keep the same score, given the strengthened protocol clarification and added analyses (failure taxonomy, parity discussion, reproducibility slice), but the key “raise score if #1/#2/#4/#7 addressed” items (granular ablations, causal attribution, scaling) are only partially addressed.

---

### Decision · Program_Chairs · 2026-01-26

Accept (Oral)